



# The day-to-day co-variability between mineral dust and cloud glaciation: A proxy for heterogeneous freezing.

Diego Villanueva[1], Bernd Heinold[1], Patric Seifert[1], Hartwig Deneke[1], Martin Radenz[1], Ina Tegen[1]

[1]Leibniz Institute for Tropospheric Research, Leipzig, 04318, Germany

*Correspondence to*: Diego Villanueva (ortiz@tropos.de)

**Abstract.** To estimate the global co-variability between mineral dust aerosol and cloud glaciation, an aerosol model reanalysis was combined with satellite retrievals of cloud thermodynamic phase. We used the CALIPSO-GOCCP and DARDAR products from the A-Train satellite constellation to assess whether clouds are composed of liquid or ice and the

MACC reanalysis to estimate the dust mixing-ratio in the atmosphere. Night-time retrievals within a temperature range from +3°C to −42°C for the period 2007-2010 were included. The results confirm that the cloud thermodynamic phase is highly dependent on temperature and latitude. However, at mid- and high latitudes, at equal temperature and within narrow constraints for humidity and static stability the average frequency of fully glaciated clouds increases by +5 to +10% for higher mineral dust mixing-ratios. The differentiation between humidity-stability regimes reduced the confounding influence

of meteorology on the observed relationship between dust and cloud ice. Furthermore, for similar mixing-ratios of mineral dust, the cloud ice occurrence-frequency in the Northern Hemisphere was found to be higher than in the Southern Hemisphere at −30°C but lower at −15°C. This may suggest a difference in the susceptibility of cloud glaciation to the presence of dust. Based on previous studies, the differences at −15°C could be explained by higher feldspar fractions in the Southern Hemisphere, while the differences at −30°C may be explained by the higher freezing efficiency of clay minerals in

the Northern Hemisphere.

## 1    Introduction

Aerosol-cloud interactions affect the Earth's climate through different mechanisms. These include impacts of aerosol particles on cloud glaciation that subsequently influence the clouds' thermodynamic phase, albedo, lifetime and precipitation. Specifically, there is growing evidence for a role of mineral dust aerosol (or of ice nucleating particles

correlated to dust aerosol) in influencing heterogeneous cloud ice formation on a global scale (Boose et al., 2016; Kanitz et al., 2011; Seifert et al., 2010; Tan et al., 2014; Vergara-Temprado et al., 2017; Zhang et al., 2018). Cloud droplets can freeze heterogeneously between 0°C and −42°C after interacting with Ice Nucleating Particles (INP) or already existing ice particles (Hoose and Möhler, 2012). It has been shown that specific aerosol types such as mineral dust and biogenic particles can act efficiently as INP already at temperatures between −10 and −20°C (Atkinson et al., 2013). Mineral dust aerosol is

emitted from arid regions, mainly from the Saharan and Asian deserts. Despite this, several dust sources exist at the Southern





mid-latitudes (e.g., Patagonia, South Africa, and Australia) and simulations show that long-range transport of dust, although sporadic, can result in considerable dust concentrations even in remote areas (Albani et al., 2012; Johnson et al., 2011; Li et al., 2008; Vergara-Temprado et al., 2017). Mineral dust aerosol is therefore suspected to be a main contributor to the atmospheric INP reservoir, especially in the Northern Hemisphere, where the mixing-ratio of dust aerosol is typically one to

two orders of magnitude larger than in the Southern Hemisphere (Vergara-Temprado et al., 2018).

The dust occurrence-frequency retrieved from spaceborne instruments like the Cloud-Aerosol Lidar with Orthogonal Polarization (CALIOP, Wu et al., 2014) has been previously used to assess the spatial correlation between dust and cloud thermodynamic phase (Choi et al., 2010; Li et al., 2017; Tan et al., 2014). Two main problems arise from this approach. First, aerosol within and below thick clouds cannot be detected by lidar. Second, low dust concentrations usually fall below

the lower detection limit of CALIOP. The Aerosol Robotic Network (AERONET, Dubovik et al., 2000), a network of ground-based remote sensing stations, has been used to evaluate and validate the dust retrievals from CALIOP. The stations from the AERONET mission use sun photometers to measure the spectrum of the solar irradiance and sky radiance to determine the atmospheric Aerosol Optical Thickness (AOT). It has been shown that the CALIOP level 2 data misses about half of the dust aerosol events detected by AERONET when the AOT is less than 0.05 (Toth et al., 2018). In contrast, dust

loadings simulated by state-of-the-art models show that most of the regions in the Southern Hemisphere have an annual mean AOT lower than 0.01 (Ridley et al., 2016).

Ice particles and cloud droplets may coexist in a so-called mixed phase state (Korolev et al., 2017). Shallow mixed-phase clouds with a liquid-dominated cloud top and ice virgae beneath are very frequent (Zhang et al., 2010), whereas cloud tops classified as mixed-phase are much rarer (Huang et al., 2015; Mülmenstädt et al., 2015). Indeed, supercooled liquid layers at

cloud top are generally observed down to temperatures of −25°C (Ansmann et al., 2008; De Boer et al., 2011; Westbrook and Illingworth, 2011). Ground-based and satellite retrievals are not yet able to accurately estimate the mass ratio of the cloud liquid and ice phase, which is especially the case in mixed-phase layers. Therefore, the Frequency Phase Ratio (FPR) is often used instead (Cesana et al., 2015; Cesana and Chepfer, 2013; Hu et al., 2010). For satellite retrievals, this is defined as the ratio of ice pixels to total cloudy pixels for a region of interest. Because most retrievals classify the cloud

thermodynamic phase either as pure ice or pure supercooled liquid, the average of the FPR represents the ratio of glaciated clouds with respect to total cloud occurrence. Therefore, the FPR should not be confused with the ice-to-liquid ratio within a single cloud. Cloud phase in the Northern and Southern Hemispheres has been studied in terms of FPR both by ground-based lidar (Kanitz et al., 2011) and by different spaceborne instruments (Choi et al., 2010; Morrison et al., 2011; Tan et al., 2014; Zhang et al., 2018). These studies found significant differences between the two hemispheres. In these studies, it has been

suggested that such differences are related to differences in aerosol and INP concentrations. Moreover, the local FPR measured at various temperatures between 3°C and −42°C by a lidar in Central Europe over a time span of 11 years has been shown to increase for higher dust loadings (Seifert et al., 2010). Furthermore, spaceborne lidar measurements of cloud thermodynamic phase and aerosol occurrence-frequency show a significant positive spatial correlation between FPR and the





frequency of detectable dust, especially at temperatures of around −20°C (Choi et al., 2010; Tan et al., 2014; Zhang et al.,
2012, 2015). This spatial correlation has been found under different atmospheric conditions including variations in humidity,
surface temperature, vertical velocity, thermal stability and zonal wind speed (Li et al., 2017a). However, the analysis of the
day-to-day variability of cloud thermodynamic phase has received less attention, especially in remote areas like the Southern
Ocean (Vergara-Temprado et al., 2017). Additionally, a more comprehensive and quantitative assessment of the potential
effect of mineral dust on cloud glaciation is currently lacking.

In this work, the role of dust aerosol on the cloud thermodynamic phase will be assessed based on the daily occurrence-
frequency of cloud glaciation around the globe between +3°C and −42°C. For this purpose, the MACC global aerosol
reanalysis will be used together with the cloud thermodynamic phase retrievals of the CALIPSO-GOCCP (Global Climate
Model Oriented Cloud Calipso Product; Cesana and Chepfer, 2013) and DARDAR-MASK (raDAR–liDAR product;
Delanoë and Hogan, 2008, 2010). The FPR obtained from these satellite products is ranked on a day-to-day basis according
to the dust mixing-ratio from the reanalysis at the moment of the retrieval, considering available observations for the period
2007-2010. Separating the retrievals in different humidity-stability regimes was crucial for assessing the real impact of dust
aerosol on cloud glaciation. This work provides a new approach to study the link between dust and cloud thermodynamic
phase variability. Its main advantage compared to previous studies is the ability to estimate aerosol mixing-ratios at cloud
level and at very low concentrations due to the use of a reanalysis dataset, which is impossible using common remote
sensing techniques, due to the detection limits of such retrievals.

In Sect. 2, the datasets used for the study and the corresponding retrieval algorithms will be presented. In Sect. 3, our
processing of the datasets will be described, including the data structure used, the different filters applied to the data and the
methodology used to assess the day-to-day correlation between dust and cloud thermodynamic phase. In Sect. 4.1, a case
study will be presented to compare the different products of cloud thermodynamic phase used in the study. The variability of
cloud thermodynamic phase with respect to latitude and temperature will be briefly assessed in Sect. 4.2 and 4.3. In Sect.
4.4, the main results will be presented showing the day-to-day correlation between dust and clouds for different latitudes,
temperature ranges, and humidity/stability regimes. In Sect. 5.1, the differences in ice occurrence-frequency between
different latitudes will be interpreted based on previous studies of the mineralogical composition of mineral dust. Finally, in
Sect. 5.2 the main assumptions, limitations and sources of uncertainty of the approach will be discussed, while conclusions
will be drawn in Sect. 6.

## 2    Data

This section presents an overview of the A-Train satellite products and the aerosol reanalysis dataset used in the study. To
focus the study of cloud phase on stratiform clouds, we will use a cloud classification product (2B-CLDCLASS) to filter out
convective clouds in Sect. 4.1-4.3. The cloud phase information will be obtained from two different products: The



CALIPSO-GOCCP product will be the focus of the study, while the DARDAR-MASK product will serve to evaluate the possible limitations of the GOCCP product. The aerosol information for the study will be obtained from the MACC reanalysis dataset and will be used in Sect. 4.4 to study the cloud phase at different mixing-ratios of mineral dust. Additionally, the ERA-Interim reanalysis will be used to obtain meteorological information and the MERRA and ECMWF-AUX reanalysis will provide the temperature profiles to rebin the satellite profiles into temperature levels.

## 2.1   2B-CLDCLASS

Different algorithms exist to classify clouds in the observations of spaceborne active instruments (Li et al., 2015). The CloudSat cloud scenario classification (2B-CLDCLASS, Sassen and Wang, 2008) mainly uses the radar reflectivity observed by the Cloud Profiling Radar (CPR) onboard CloudSat together with the attenuated backscatter signal from CALIOP to classify clouds into 8 different types (Sassen and Wang, 2008). These are: low-level (stratocumulus and stratus), mid-level (altostratus and altocumulus) and high-level clouds (cirrus), and clouds with vertical development (deep convection clouds, cumulus, and nimbostratus). The main criteria for the classification of non-precipitating clouds are the radar reflectivity and temperature obtained from the ECMWF-AUX product. The CPR is mainly sensitive to large particles (e.g., raindrops) and therefore clouds with a reflectivity larger than a given temperature-dependent threshold can be defined as precipitating. The fifth range gate of the CPR (~1.2 km above ground level) is used for this classification. The threshold is a function of temperature, and ranges from -10 to 0 dBZ (Hudak et al., 2009). The standard error of the ECMWF-AUX temperature, which is based on the Integrated Forecast System of the European Center for Medium Range Weather Forecasting (ECMWF), has been estimated to be around 0.6 K in the troposphere (Benedetti, 2005).

## 2.2   CALIPSO-GOCCP

The CALIPSO-GOCCP v.3.0 product (Cesana and Chepfer, 2013) uses the Attenuated Total Backscatter (ATB), the molecular ATB ($ATB_{mol}$) and the cross-polarized ATB ($ATB_{\perp}$) from CALIOP at 532 nm wavelength to detect cloudy pixels. The nadir angle of CALIOP was increased from 0.3° to 3° in November 2007 to reduce specular returns from horizontally oriented ice crystals. The lidar has a horizontal resolution of 333 m and a vertical resolution of 30 m. Cloudy pixels are defined as pixels with a scattering ratio higher than 5 ($SR = ATB/ATB_{mol} > 5$). The cloud volume fraction is defined as the ratio of cloudy to total pixels within a gridbox. The product also uses the depolarization ratio of the retrieved signal components to make a decision on cloud-phase (ICE or LIQUID). The decision is based on an empirical threshold for the depolarization ratio of ice particles and is made for each pixel, with a a vertical resolution of 480 m. From this information, the FPR is calculated for each 2°×2° gridbox. These gridboxes are then regridded into 3 K bins using the temperature from the Modern Era Retrospective-analysis for Research and Applications (MERRA, Bosilovich et al., 2011) reanalysis.





## 2.3    DARDAR-MASK

The DARDAR-MASK v1.1.4 product (Delanoë and Hogan, 2008, 2010) available at the ICARE data center combines the attenuated backscatter from CALIOP (at 532 nm; sensible to small droplets), the reflectivity from the CPR (at 94 GHz; sensible to larger particles) and the temperature from the ECMWF-AUX product to assess cloud thermodynamic phase. The radar pixels have a horizontal resolution of 1.4 km (cross-track) $\times$ 3.5 km (along-track) and a vertical resolution of 500 m, with a nadir angle of 0.16° of the radar beam. A decision is made for each pixel with a 60 m vertical resolution to take

advantage of the lidar resolution. These pixels are collocated with the CloudSat footprints (1.1 km horizontal resolution). If the backscatter lidar signal is high (>2·10$^{-5}$ m$^{-1}$ sr$^{-1}$), strongly attenuated (down to at least 10% in the next 480 m) and penetrates less than 300 m into the cloud, it is assumed that supercooled droplets are present. In this case, the pixel is categorized as supercooled or mixed-phase depending on the radar signal, which is assumed a priori to indicate the presence of ice particles. Otherwise, the pixel is categorized as ice (Delanoë et al., 2013; Mioche et al., 2014). For reasons that will

become clear later, we will coerce the mixed-phase category into the liquid category.

## 2.4    MACC and ERA-Interim

The Monitoring Atmospheric Composition and Climate reanalysis (MACC, Eskes et al., 2015) is based on ECMWF's Integrated Forecasting System (IFS), and simulates the emission, transport and deposition of various aerosol species and trace gases with an output resolution of 1.125° $\times$ 1.125° and 60 vertical levels. In this study, we use the dust mixing-ratio

and large-scale vertical velocity from the daily MACC reanalysis product on model levels provided by the ECMWF. Additionally, the Relative Humidity (RH) from the ERA-Interim reanalysis daily product (Dee et al., 2011) will be used in Sect. 5. The cloud properties in the MACC reanalysis are derived from the ECMWF Integrated forecast system (IFS Cycle 36r1 4D-Var). This atmospheric model is analogous to the one used in the ERA-Interim reanalysis (IFS Cycle 31r2 4D-Var). At the time of this study, the new generation of reanalysis based on IFS Cycle 41r was not yet publicly available. However,

it is expected that future studies will use the new CAMS (Copernicus Atmosphere Monitoring Service) and ERA5 reanalysis instead of the MACC and Era-Interim reanalysis.

Dust emission in the MACC model is parameterized as a function of the 10 m wind, vegetation, soil moisture and surface albedo. The dust loadings are corrected by the assimilation of the total column AOT at 550 nm retrieved from the MODIS instrument on board NASA's Aqua and Terra satellites. Dust sinks are simulated including dry and wet deposition, as well as

in-cloud and below-cloud removal.

The freezing efficiency of INPs depends mainly on their surface area concentration (Atkinson et al., 2013; Hartmann et al., 2016; Murray et al., 2011; Niedermeier et al., 2011, 2015; Price et al., 2018). While the number concentration of dust aerosol is generally dominated by fine-mode dust (particle diameter < 0.5 μm), the surface area concentration is often determined by both fine and coarse (particle diameter > 1 μm) dust particles (Mahowald et al., 2014). Moreover, the atmospheric lifetime of



fine-mode dust is longer than that of coarse-mode dust due to the lower dry deposition rates of finer particles (Mahowald et al., 2014; Seinfeld and Pandis, 1998). Additionally, it has been shown that the MACC model underestimates the coarse-mode dust fraction in relation to fine-mode dust (Ansmann et al., 2017; Kok, 2011). In the MACC reanalysis, dust aerosols are represented by three size bins, with size limits of 0.03, 0.55, 0.9 and 20 μm diameter. In this work, we will define the size bin between 0.03 and 0.55 μm as *fine-mode dust*. Because the fine mode contributes to both the number and surface area

concentration it will be used as a proxy for the concentration of dust INP. Although mostly focused on the Northern Hemisphere, several studies have evaluated the simulated dust mixing-ratios from the MACC reanalysis with observations. A mean bias of 25% was found between MACC and LIVAS, a dust product based on CALIPSO observations over Europe, northern Africa and Middle East (Georgoulias et al., 2018). Additionally, the correlation between MACC and AERONET was found to range from 0.6 over the Sahara and Sahel to 0.8 over typical regions of dust transport (Cuevas et al., 2015).

Finally, using shipborne measurements of long-range dust transport, it was found that the MACC model significantly overestimates the fine-dust fraction compared to observations (Ansmann et al., 2017).

## 3    Methods

In this section, the different steps in our processing of the datasets presented in Sect. 2 will be described. **Fig. 1** presents a flow chart of the data processing and a roadmap for the following subsections.

### 3.1    Selection of cloud profiles

To exclude the effects of the scattering of sunlight on the detection of the CALIOP lidar signal, only night-time retrievals were used. Additionally, to avoid biases in the radar retrievals at pixels where the lidar is fully attenuated only pixels where the CALIOP retrieval was classified as cloudy (SR > 5) were used. This warranted a dataset free of lidar-attenuated pixels. To avoid biases on the radar reflectivity due to rain droplets, only non-precipitating clouds were included in Sect. 4.1-4.3.

Using the 2B-CLDCLASS classification, we defined non-precipitating gridboxes (1.875x1.875) as containing less than 10 % precipitating pixels compared to the total number of cloudy pixels. The three filters (night-time, lidar-not-fully-attenuated and non-precipitating) were applied to both the CALIPSO-GOCCP and DARDAR-MASK cloud products.

### 3.2    Regridding and rebinning: Temperature levels and 1.875°×30° gridboxes

It will become clear in Sect. 4.2. That the cloud thermodynamic phase is mainly a function of temperature. Anticipating this,

temperature bins of 3 K each were used as vertical coordinate throughout the study. The temperature profiles were obtained from the ECMWF-AUX reanalysis for the DARDAR and CLDCLASS products and from the MERRA reanalysis for the GOCCP product.


To fill the horizontal gaps between the satellite orbits, we regridded the dataset into a Gaussian T63 grid, aggregating 16 gridboxes along the longitude (1.875°×30°; lat×lon). The Gaussian T63 grid is commonly used in Global Climate Models
(Randall et al., 2007) and facilitates future comparisons with global simulations of cloud thermodynamic phase. In section 4.4 and onwards, latitude bands of 30° are used to allow a direct comparison with previous studies (Zhang et al., 2018).

### 3.3 Frequency Phase Ratio and cloud volume weighting

Huang et al. (2015) compared the cloud thermodynamic phase retrieved by the DARDAR-MASK and the CALIOP level 2 cloud layer product (Hu et al., 2009) for clouds over the Southern Ocean (at 40-60°S, 125-145°W) and over the North
Atlantic (45°-65°N, 13°-35°W). In this study, at a cloud top temperature of −10°C almost all cloud tops are classified as liquid by the CALIOP product, whereas most clouds are classified as ice or mixed-phase by the DARDAR-MASK. This discrepancy is attributed partly to the instrumental differences and partly to differences in the classification algorithms.

To assess the differences between the cloud phase from the DARDAR-MASK and CALIPSO-GOCCP products, we defined a new phase ratio based on the DARDAR-MASK classification. In this alternative definition, which we will call ALT-
DARDAR, only gridboxes (1.875°×30°×3 K) *filled* with ice pixels are considered as ice (*fully* glaciated), so that just a single liquid pixel is enough to define a gridbox as liquid (not *fully* glaciated). One advantage of this marginal definition is that it ignores cloud ice in mixed-phase clouds, which is mostly only detected as such by the DARDAR-MASK product and neglected by the CALIPSO-GOCCP product.

For $FPR_{GOCCP}$ and $FPR_{DARDAR}$, the FPR is calculated as the ratio of ice pixels to the total number of pixels within each
gridbox. The $FPR\_ALT_{DARDAR}$ uses gridboxes instead: The gridbox phase is set to one (*fully* glaciated cloud) if all cloud pixels in the gridbox are classified as ice and zero (not *fully* glaciated cloud) otherwise. The differences in $FPR_{GOCCP}$, $FPR_{DARDAR}$ and $FPR\_ALT_{DARDAR}$ will be studied in detail in Sect. 4.1-4.2. Here it will be shown that the $FPR\_ALT_{DARDAR}$ definition does well in mimicking the limitations of the CALIPSO-GOCCP product.

We expect cloudy overcast gridboxes to be statistically more robust than partly cloudy gridboxes. Cloud cover is generally
defined for the whole atmosphere or certain levels (low, mid or high). Therefore, we use instead the cloud volume fraction (also known as three-dimensional or 3-D cloud fraction; Chepfer et al., 2010, 2013; Li et al., 2017a; Yin et al., 2015) retrieved by the CALIPSO-GOCCP product as a weight for the averages used in Sect. 4.1-4.3. The cloud volume fraction is defined as the number of cloudy pixels divided by the total number of pixels within a given length-height domain along the satellite swath. The length is the segment of the satellite swath crossing a given gridbox, and the height interval corresponds
to each temperature bin (3 K) in this study. The main benefit from using the cloud volume fraction instead of the cloud cover is that the former is defined for each temperature bin. This allows to differentiate between vertically thick and shallow clouds. Using the cloud volume fraction as a weight results in a higher representation of clouds with larger spatial extension (vertical as well as horizontal). It also introduces a bias towards the cloud tops for thick clouds because the lidar signal is



attenuated at higher cloud depths. More details on the spatiotemporal variability of the cloud volume fraction can be found
on the supplement (S8) to this article.

In Sect. 4.1, the adjusted ice volume fraction

$$FPR^* = (2 \cdot FPR - 1) \cdot cvf \tag{3.1}$$

is used instead of the traditional *FPR,* with *cvf* the cloud volume fraction obtained from the GOCCP product. The adjusted
FPR* helps to visualize the cloud thermodynamic phase of the significant (high *cvf*) clouds in the retrieval. In Sect 4.2-4.3,
the FPR averages (for each dimension) were calculated as

$$FPR_{avg} = \frac{\sum(cvf_i \cdot FPR_i)}{\sum cvf_i} \tag{3.2}$$

with *cvf$_i$* the stratiform cloud volume fraction in each gridbox, defined using the 2B-CLDCLASS classification as

$$cvf_i = cvf_{i,altostratus} + cvf_{i,cirrus} + cvf_{i,altocumulus} + cvf_{i,stratocumulus} \tag{3.3}$$

### 3.4   Meteorological regimes

Dust aerosol can produce or be accompanied by changes in atmospheric stability and relative humidity. To disentangle such
effects, we constrain the cloud environment in Sect. 4.4 using the air relative humidity, the lower troposphere static stability
(LTSS) and the upper tropospheric static stability (UTSS). The latter two are defined as:

$$LTSS = T_{700} \left[\frac{1000}{700}\right]^{\frac{R}{Cp}} - T_{sfc} \left[\frac{1000}{p_{sfc}}\right]^{\frac{R}{Cp}} \tag{3.4}$$

$$UTSS = T_{350} \left[\frac{1000}{350}\right]^{\frac{R}{Cp}} - T_{500} \left[\frac{1000}{500}\right]^{\frac{R}{Cp}} \tag{3.5}$$

With $T_x$ and $P_x$ the temperature and pressure at the surface or at *x* hPa using the pressure levels of the ERA-Interim
reanalysis. $R$ is the gas constant and $C_p$ the specific heat capacity of air (Klein and Hartmann, 1993). The relative humidity is
obtained directly from the ECMWF-AUX reanalysis.

To increase the sample size prior to the regime classification, we included back gridboxes containing precipitating or
convective clouds back into the dataset. However, most of such clouds are expected to fall into high RH and low LTSS
regimes and, therefore, could still be excluded later on.

### 3.5   Classification of dust loads and day-to-day correlation

In contrast to previous studies, in this work we want to isolate the day-to-day correlation between dust aerosol and cloud
phase. To exclude the spatial component of the correlation, the complete time span 2007-2010 is used to determine the time-
deciles of dust mixing-ratio using the MACC reanalysis for each volume gridbox (latitude, longitude, temperature). These
deciles are used to sort the daily data depending on the daily dust mixing-ratio into 10 different decile ranks. These ranks can
be also understood as dust mixing-ratio bins (from now on simply *deciles*).





Next, we also need to exclude the seasonal component of the temporal correlation. With this purpose, for each 3 K temperature bin and each gridbox the daily data is averaged within each dust decile and each month of the year. This is done as a multiyear average (e.g., January containing Jan'07, Jan'08, Jan'09 and Jan'10). The resulting field contains one extra

dimension for each gridbox (month, *dust decile*, temperature, latitude, longitude). **Fig. 2** present a visualization of this process.

### 3.6    Data availability and averaging order

**Fig. 3** shows the zonal sum of the sample size for the FPR$_{GOCCP}$ at −15°C and −30°C. Each count corresponds to a month-decile pair. Within a 12 K range, each latitude bin (1.875°) contains about 1500 to 2000 observational datapoints in the mid-

latitudes and about 500 to 1500 datapoints in the high-latitudes, with the lowest sample size found for the high southern latitudes. Potential reasons for missing data are:

- The satellite swaths (orbits) produce a different density of retrieved profiles at different latitudes.
- Using only night-time data, the sample size in the meteorological summer time (shorter nights) is lower.
- The cloud phase retrievals are less frequent for seasons, regions and heights with low cloud cover.
- At high latitudes, relatively warm temperatures (e.g., -15°C) exceeding the surface temperature can be found, and therefore no information is available for such temperatures (e.g., over Antarctica in winter).

To avoid artefacts arising from the averaging of dimensions containing missing values, the averaging order of dimensions was defined (going from the first to the last dimension to be averaged) as: longitude, month, decile, latitude, temperature. Latitude and temperature are averaged last because of the higher associated correlations with cloud phase (Sect. 4.2 - 4.3 of

this study; Choi et al., 2010; Tan et al., 2014). Each 1.875°×30° gridbox contains on average 100 to 200 datapoints at −15°C (within a 12 K range) in the mid-latitudes. Meanwhile, in the subtropics and in the high latitudes, the sample size is much more heterogeneously distributed and can drop below 50 datapoints near the poles and in subsidence regions. A detailed view of the spatiotemporal distribution of the sample size for stratiform clouds can be found in the supplement (S14) to this article.

## 4    Results

### 4.1    Case study

**Fig. 4** shows a case study at 9:50 UTC on Dec 14, 2010 over the Southern Ocean for temperatures between −42°C and +3°C. This A-train segment has been already chosen for a previous case study by Huang et al. (2015) due to the variety of cloud types it contains. **Fig. 4a-b** show for the same segment the cloud volume cover (CALIPSO-GOCCP) of clouds classified

(2B-CLDCLASS) as cirrus or altocumulus (**Fig. 4a**) and as altostratus or stratocumulus (**Fig. 4b**). These cloud types are frequently thin enough to be penetrated by lidar and radar systems and are therefore a good target to study cloud glaciation





processes (Bühl et al., 2016; D.Zhang et al., 2010b). **Fig. 4c** shows the mixing-ratio of fine (0.03µm-0.55µm) dust aerosol (MACC reanalysis) for the same vertical plane. **Fig. 4d-f** shows the FPR* (see Sect. 3) which is weighted by cloud volume fraction to highlight the phase of extensive clouds.

Some major differences can be observed between the three FPR* variables in **Fig. 4d-f**. For the altocumulus cloud at 35-40°S and +3°C to −6°C, the ice virgae falling from the cloud (FPR_{DARDAR}) are missed in the FPR_{GOCCP}. Because this study aims at assessing the occurrence-frequency of fully glaciated clouds, such mixed-phase clouds are then reclassified in FPR_ALT_{DARDAR} as liquid clouds. A similar case is observed for the stratocumulus clouds at 50-55°S and +3°C to −6°C, and for the altostratus at 35-45°S below the −20°C isotherm (at higher temperatures). Finally, the cirrus clouds above −33°C

remain nearly unaffected by the reclassification in FPR_ALT_{DARDAR} as it is classified as fully glaciated. Clouds between -38°N and -44°N, ranging from -6°C to -33°C in temperature, are classified mostly as *altostratus* by the 2B-CLDCLASS product. These altostratus clouds offer a good opportunity to compare the three FPR variables in more detail.

*FPR_{GOCCP}*: The detected ice virgae below the liquid cloud top suggest that the cloud top did not fully attenuate the lidar signal (not optically thick enough). The number and/or size of the ice particles near the cloud top probably was not enough to

increase the depolarization ratio above the threshold value for the GOCCP algorithm and was therefore classified as liquid.

*FPR_{DARDAR}*: In the decision tree of the DARDAR algorithm there are multiple alternatives for a mixture of cloud droplets and ice particles (e.g., at cloud top) to be classified as ice only (Mioche et al., 2014):

   a)   If the lidar backscatter signal (β) is lower than $2.10^{-5}$ m$^{-1}$ sr$^{-1}$

   b)   If not *a)*: If it is *weakly attenuated* (less than 10 times) or *not rapidly attenuated* (at a depth larger than 480 m).

c)   If not *b)*: If the *layer thickness* of the cloud is larger than 300 m. This is equivalent to 5 pixels with a lidar vertical resolution of 60 m.

Therefore, there are many cases where a mixed-phase cloud (and especially an optically thin stratiform cloud) can be miss-classified as ice only in the DARDAR product and consequently in the FPR_DARDAR variable. In this specific case, we speculate that *c)* is the most probable cause because of the large vertical extent of the clouds around 1 to 5 km using a moist

adiabatic lapse rate of −6 K/km for the estimation).

*FPR_ALT_{DARDAR}*: In the case of droplets and ice particles coexisting at cloud top, we expect that at some location the cloud droplets will be enough in number for the pixel to be classified as liquid (strong attenuation) in the DARDAR-MASK algorithm. If this is the case, the entire gridbox value of FPR_ALT _{DARDAR} will be LIQUID. This should be interpreted as a *non-completely glaciated* cloud.

In summary, the GOCCP algorithm is unable to detect ice in mixed-phase clouds and the DARDAR algorithm tends to classify mixed-phase clouds as ice. Therefore, we avoid using the *frequency of cloud ice* (FPR) to compare the GOCCP and DARDAR products. Instead, we introduced FPR_ALT_{DARDAR}, which has been defined to address the limitations of both products. In FPR_ALT_{DARDAR}, a significant portion of mixed-phase clouds that would otherwise be classified as ICE are now classified as LIQUID. This however partly reintroduces the inability of the GOCCP algorithm to detect ice in mixed-phase





clouds. Therefore, the *frequency of **completely glaciated** clouds,* which is represented by FPR_ALT$_{DARDAR}$ and FPR$_{GOCCP}$,
allows a better comparison of both algorithms, mostly by ignoring ice virgae in FPR_ALT$_{DARDAR}$ when cloud droplets are
also present in the same gridbox. This idea is summarized in Table 1.

### 4.2    Temperature dependence

Mixed-phase clouds between 0°C and −25°C are usually topped by a liquid layer (Ansmann et al., 2008; De Boer et al.,
2011; Westbrook and Heymsfield, 2011). Below this layer, there is often a thicker layer containing ice particles. Because the
CPR is more sensitive to larger particles, this results in a large fraction of the cloud classified as ICE in the DARDAR-
MASK. In contrast, the CALIOP backscatter signal is usually already strongly attenuated at such depths and often cannot
detect large ice particles. Therefore, the CALIPSO-GOCCP algorithm usually classifies the whole cloud layer as liquid
(Huang et al., 2012; 2015). As a result, FPR$_{DARDAR}$ tends to be higher than FPR$_{GOCCP}$.

**Fig. 5** shows that the global average FPR$_{GOCCP}$ as a function of temperature decreases roughly from 100 % at −40.5°C to
about 20 % at −1.5°C and down to 0 % at +1.5°C. This temperature dependence between −42°C and 0°C is also observed for
a wide range of parameterizations in global climate models (Cesana et al., 2015), in ground-based measurements (Kanitz et
al., 2011), in spaceborne lidar measurements (Tan et al., 2014) and in aircraft measurements (McCoy et al., 2016). However,
for the same temperature range, the FPR$_{DARDAR}$ only decreases down to 60 % at 1.5°C. This is partly due to the higher
sensitivity of the radar to ice particles, especially falling ice. Additionally, in the DARDAR algorithm water can be still
classified as ice at +1.5°C due to the melting layer being set to a wet-bulb temperature (Tw) of 0°C. This allows the
detection of ice at temperatures slightly above 0°C dry-bulb temperatures (named simply temperature in this work). For
instance, at a relative humidity of 50%, a temperature of about +2.5°C would correspond to a Tw of −2.5°C. Nevertheless,
this last effect is not relevant for temperatures below freezing.  In contrast, FPR_ALT$_{DARDAR}$ follows very closely the pattern
of the FPR$_{GOCCP}$ down to −1.5°C. The absolute differences of the global averaged FPR_ALT$_{DARDAR}$ and FPR$_{GOCCP}$ are less
than 10 % between −42°C and 0°C. This shows that the temperature dependence of the alternative phase ratio
FPR_ALT$_{DARDAR}$ and FPR$_{GOCCP}$ agree better than for FPR$_{DARDAR}$. Therefore, for the rest of the study, only FPR_ALT$_{DARDAR}$
and FPR$_{GOCCP}$ will be considered.

Additionally, the average fine-mode dust mixing-ratio is also shown in **Fig. 5**. At 0°C the mixing-ratio is five times higher
than at −42°C (note the logarithmic right y-axis). This reflects the fact that dust mixing-ratios tend to be lower at higher
altitudes where temperatures are lower. However, there are important exceptions to this, such as in the long-range transport
of dust layers over the ocean.

### 4.3    Latitude dependence

**Fig. 6** shows the latitudinal dependence of dust and cloud thermodynamic phase at −30°C (averaged from −36°C to −24°C;
Fig5a) and at −15°C (averaged from −21°C to -9°C; Fig 5b). For both temperature ranges shown in **Fig. 6** the absolute





maximum of FPR is located near the Equator. At −30°C, the maximum is 85% for FPR$_{GOCCP}$ and 78% for FPR_ALT$_{DARDAR}$ and at −15°C the FPR from both products peak at 44%. These maxima are probably associated with the enhanced homogeneous freezing in the tropics at temperatures below −40°C and the resulting downward transport of cloud ice. Similarly, minima are observed towards the high latitudes. At −30°C, the FPR$_{GOCCP}$ has two local maxima with values of 76

% and 84 % near 39°S and 39°N, respectively. Similar local maxima are observed for the FPR_ALT$_{DARDAR}$ but at higher latitudes, at 61°S and 61°N with values 69 % and 74 %. At −30°C, both products show a higher FPR in the Northern Hemisphere than in the Southern Hemisphere, in particular for the high latitudes. This higher FPR coincides with the higher average dust mixing-ratio in the Northern Hemisphere. Such positive spatial correlations between FPR and dust aerosol have been already pointed out using the dust occurrence-frequency derived from CALIOP (Choi et al., 2010; Tan et al., 2014;

Zhang et al., 2012).

In comparison, the differences between FPR$_{GOCCP}$ and FPR_ALT$_{DARDAR}$ at −15°C are much lower than at −30°C as shown in **Fig. 6b**. Moreover, the FPR$_{GOCCP}$ at −15°C is lower than the FPR_ALT$_{DARDAR}$ at the southern mid-latitudes and northern high-latitudes. For both variables, a local minimum near 73°S is followed by a steep increase at 84°S. The larger standard deviation in these latitudes is possibly a result of the low sample size in this region, as mentioned in Sect. 2. However, the

higher FPR in the southern than in the northern polar region is consistent with the fraction of ice clouds reported previously in the literature at −20°C (Li et al., 2017). The predominance of ice clouds in Antarctica has been already pointed out earlier in the literature (Ardon-Dryer et al., 2011; Bromwich et al., 2012). Incoming air masses from the ocean may carry higher concentrations of INP like biogenic aerosol (Saxena, 1983), Patagonian soil dust or Australian black carbon (Bromwich et al., 2012). Particularly, immersion freezing INP are thought to be significant in Antarctica (Ardon-Dryer et al., 2011;

Bromwich et al., 2012). Similarly, it has been shown that the orographic forcing in Antarctica can lead to high ice water contents for maritime air intrusions (Scott and Lubin, 2016). In other words, maritime air intrusions associated with higher temperatures, higher concentrations of INP and stronger vertical motions could explain the observed pattern in the southern polar regions.

The pattern of the mean large-scale vertical velocity (MACC reanalysis) of the clouds studied is particularly similar to the

FPR at −15°C. Moreover, the spatial correlation between large-scale updraft velocity at 500 hPa is positively correlated to the occurrence-frequency of ice clouds at −20°C (Li et al., 2017a). In other words, both the dust mixing-ratio and the large-scale vertical velocity seem to be positively correlated (spatially) to FPR. There are some plausible explanations for this:

- The spatial correlation can be a result of an enhanced transport of water vapor to higher levels at temperatures below −40°C and the subsequent sedimentation of ice crystals from the homogeneous regime (cloud seeding).

- The updrafts are associated with higher availability of INP at the cloud level (from below the cloud) and the effect is large enough to mask the enhanced droplet growth typically associated with updrafts.





- The updrafts enhance a certain type of heterogeneous nucleation requiring saturation over liquid water (e.g., immersion freezing). Updrafts generate a local adiabatic cooling, possibly activating INPs that may have not been active before at higher temperatures.

However, to understand which (if any) of these explanations influence the freezing processes inside the cloud remains a complex challenge for ongoing debate (Sullivan et al., 2016).

### 4.4  Constraining the influence of static stability and humidity on the dust-cloud-phase relationship

To study the temporal correlation between mineral dust mixing-ratio and cloud ice occurrence-frequency (from now on referred to as *dust-cloud-phase relationship*) it is crucial to systematically classify weather regimes to constrain the 375  meteorological influence. By doing so, the resulting dust-cloud-phase relationship may offer a good insight into how heterogeneous nucleation of mineral dust may affect the day-to-day average cloud thermodynamic phase. In other words, to extract the specific influence of mineral dust on cloud glaciation, it is necessary to identify and constrain relevant meteorological confounding factors (Gryspeerdt et al., 2016). The atmospheric relative humidity and static stability are good candidates for such a confounding factor (Zamora et al., 2018). Both are correlated with the transport of mineral dust and 380  vary between different cloud regimes. Additionally, relative humidity is, next to the temperature, one of the main factors in the initiation of ice nucleation in laboratory studies (Hoose and Möhler, 2012; Welti et al., 2009). The static stability of the atmosphere (see equations 3.4 and 3.5) represents the gravitational resistance of an atmospheric column to vertical motions and is defined as the difference of potential temperature between two pressure levels (Klein and Hartmann, 1993). Because such vertical motions are traduced in a temperature change rate within the air parcel, the static stability can have an 385  important impact on the heterogeneous freezing rates, especially on immersion freezing. We note that the dynamic component of the atmospheric stability is not included in the static stability. Especially in the upper troposphere, atmospheric gravity waves occurring during stable thermal conditions may also result in vertical motions affecting ice production.

In general, moist and unstable conditions are associated with enhanced lifting of air that likely causes nucleation of hydrometeors. The effect of humidity and static stability on ice production is not straightforward. In moist convective 390  conditions (high humidity and low static stability) between 0°C and -40°C, the supersaturation of water vapor over liquid increases the liquid formation because ice growing (deposition nucleation) is rather inefficient at this conditions. However, at temperatures below −40°C, ice production by deposition and homogeneous nucleation are favoured, which can result in a higher occurrence of cloud ice in the mixed-phase regime below due to ice sedimentation. To constrain both the atmospheric stability and humidity, a subset of the data must be found within a narrow range of these variables. At the same time, enough 395  data points must still be available to assess the dust-cloud-phase relationship. For this purpose, we use a probability histogram to define the regime bounds such that at least 10% of the data is included in each regime.

**Fig. 7** shows the probability density function of the dataset against the relative humidity from the ECMWF-AUX dataset and the static stability from the ERA-Interim reanalysis at −22°C. The bounds for each regime are shown with boxes. For the





relative humidity, the bounds are defined at 60, 70 and 80%, for the LTSS at 10, 15 and 20 K, and for the UTSS at 4, 6 and

8 K. The fraction of data inside each regime correspond to the integral of the probability density within the regime bounds. For example, if the probability density between 4−6 K and 70−80% is 0.01, then 20% of the data is contained between these bounds. The magenta boxes represent the different stability-humidity regimes used for the lower and upper troposphere. To maximize the sample size, for this classification and for the following results precipitating and convective clouds are also included.

**Fig. 8** shows the dust-cloud-phase relationship for the mid- and high-latitudes separated by humidity and LTSS at −15°C using the FPR$_{GOCCP}$ product and MACC reanalysis. For dust mixing-ratios between 0.1 and 2.0 µg kg$^{−1}$, the cloud-phase curve in the mid-latitudes follows a similar logarithmic increase of cloud ice occurrence-frequency of about +6% for low-LTSS and +4% for high-LTSS conditions. After analysing 11 years of ground-based lidar measurements in Leipzig, Seifert et al. (2010) reported a slightly higher increase by about +10% between −10°C and −20°C for dust concentrations

between 0.001 to 2 µg m$^{−3}$ (note the different units). In our results at −15°C, the cloud ice occurrence-frequency tends to be higher for higher relative humidity and the LTSS seems to have a major effect on the dust-cloud-phase relationship. For high-LTSS conditions **(Fig. 8a-b)**, a positive dust-cloud-phase correlation can be observed at all four latitude bands. The slope is similar for the Northern and Southern Hemisphere and for mid- and high latitudes. Because the horizontal axis is logarithmic, this means that for the high-latitudes small increases in dust mixing-ratio are associated with a high increase in

cloud ice occurrence-frequency. However, the range of ice occurrence-frequency values is higher for the high latitudes. Particularly for the low-RH regime, the ice occurrence-frequency in the southern high latitudes increases by +8%, and at the mid-latitudes, the increase is only about +4%. In both mid- and high latitudes, the cloud ice occurrence-frequency for the same dust mixing-ratio is about +2% to +8% higher in the Southern than in the Northern Hemisphere. This could point to a factor other than dust aerosol causing an increased ice occurrence-frequency in the Southern Hemisphere. It could also

suggest a potential difference in the sensitivity of cloud glaciation to mineral dust between hemispheres. The difference between the Northern and Southern Hemisphere is reduced in the high-RH regime, as well as the standard deviation of the ice occurrence-frequency, possibly due to the higher sample size density in the high-RH regime. For the low-LTSS regime **(Fig. 8c-d)**, the cloud thermodynamic phase in the high-latitudes remains mostly constant for increasing dust mixing-ratios, and the dust-cloud-phase curves for the mid-latitudes coincide so that the maximum cloud ice occurrence-frequency in the

southern mid-latitudes is similar to the minimum in the northern mid-latitudes.

**Fig. 9** show a similar constraint on humidity and UTSS at −30°C. For all regimes, the cloud ice occurrence-frequency in the southern high latitudes remains almost constant for increasing dust mixing-ratios. For the high-RH regime, the cloud ice occurrence-frequency tends to be higher, especially for the southern high latitudes for which the cloud ice occurrence-frequency is about +4% higher at the high-RH regime. For dust mixing ratios between 0.1 and 1.5 µg kg$^{−1}$, the cloud ice

occurrence-frequency at −30°C increase by about +5%. The highest increase is found for the northern latitudes. However, the results from the southern mid-latitudes contradict the notion that the INP activity of mineral dust is of secondary



importance in the Southern Hemisphere due to low dust aerosol concentrations (Burrows et al., 2013; Kanitz et al., 2011). Nevertheless, recent studies have acknowledged that the importance of mineral dust in the southern latitudes still cannot be ruled out (Vergara-Temprado et al., 2017). **Fig. 10** shows the dust-cloud-phase relationship for different UTSS and relative

humidity at −22°C. Similar to the results at −15°C and −30°C, the cloud ice occurrence-frequency is higher in the high-RH regime. For high-UTSS conditions, the dust-cloud-phase curves are in closer agreement between the Northern and Southern Hemisphere. Overall, at −22°C the four latitude bands show the best agreement between Northern/Southern Hemisphere and mid-/high- latitudes. Combining the results from all mid- and high latitudes, the ice occurrence-frequency increases by about 25% at high-UTSS conditions and by about 20% at low-UTSS conditions for mixing ratios between 0.01 and 1.0 µg kg$^{-1}$ at

−22°C. This suggests that the dust mixing-ratio may explain both the day-to-day differences in cloud ice occurrence-frequency and the differences between latitudes.

In general, for temperatures between −36°C and −9°C, higher fine-mode dust mixing-ratios are associated with an increasing cloud ice occurrence-frequency. The results suggest that only the lower static stability at −15°C has a strong influence on the relationship between mineral dust and cloud ice. This is may be a consequence of the dynamic component of the atmospheric

stability at lower temperatures (e.g., gravity waves), which is not included in the static stability parameter. At all temperatures studied, higher humidity values were associated with a higher cloud ice occurrence-frequency. For similar dust loadings, the cloud ice occurrence-frequency was found to be higher at the mid-latitudes than at the high-latitudes. However, against our expectations, for similar dust loadings the cloud ice occurrence-frequency at −15°C was higher in the Southern than in the Northern Hemisphere.

**5    Discussion**

Some studies have already suggested that the lower occurrence-frequency of cloud ice in the higher latitudes may be associated with lower INP concentrations (Li et al., 2017a; Tan et al., 2014; Zhang et al., 2012). This hypothesis has been supported mainly by the spatial correlation between the dust relative aerosol frequency and the occurrence-frequency of ice clouds retrieved from satellite observations. However, evidence of the global temporal co-variability between INP and ice

occurrence-frequency on a day-to-day basis was lacking up to now. Furthermore, by studying the temporal correlation between mineral dust and cloud ice occurrence-frequency it is possible to extract new information about the differences in cloud glaciation at different latitudes and to connect these differences to previous studies of heterogeneous freezing. Particularly, our results may be used to evaluate our current knowledge of the global differences in the mineralogy of dust aerosol and its freezing efficiency.





## 5.1 North-South contrast

We have found that the ice occurrence-frequency can vary at different latitudes even for similar mixing-ratios of mineral dust. This could be explained by differences in the mineralogical composition of the mineral dust aerosol at the Southern and Northern Hemisphere. It has been suggested that the freezing efficiency of clay minerals from the Northern Hemispheres (composed mostly from Illite and Smectite minerals) can be well represented by the mineral Montmorillonite while the Southern clay minerals are better represented by the mineral Kaolinite (Claquin et al., 1999; Hoose et al., 2008). The freezing efficiency of Kaolinite and Montmorillonite are known for the immersion and contact freezing mode (Diehl et al., 2006; Diehl & Wurzler, 2004). Following this simplification, the immersion freezing rates at −30°C would be 300 times higher in the Northern than in the Southern Hemisphere. This could explain the higher ice occurrence-frequency in the Northern Hemisphere relative to the Southern Hemisphere for similar dust mixing-ratios at −30°C. Below −25°C, the contact freezing is expected to dominate over immersion freezing. However, for contact freezing between −25°C and −16°C the freezing rate is similar for Kaolinite and Montmorillonite. This again may explain why the ice occurrence-frequency in the Northern Hemisphere is only slightly higher for similar dust mixing-ratios at −22°C. Finally, between −15°C and −4°C, the contact freezing efficiency of Montmorillonite is slightly higher than for Kaolinite. However, this fails to explain the higher ice occurrence-frequency found in the Southern Hemispheres at −15°C. Nevertheless, at such high temperatures, other dust minerals like feldspar mineral are much more efficient as ice nucleating particles than clay minerals (Atkinson et al., 2013). Moreover, it could be that the effect of such feldspar minerals dominates over the effect of clay minerals at high temperatures. Indeed, such efficient minerals are believed to be quickly depleted trough heterogeneous freezing, so that only few would reach lower temperatures. Therefore, they are likely more relevant at temperatures above −20°C, where the immersion efficiency of clay minerals quickly decay (Boose et al., 2016; Broadley et al., 2012; Murray et al., 2011). If feldspar minerals do dominate the heterogeneous freezing due to mineral dust above −20°C, then the higher cloud ice occurrence-frequency in the Southern Hemisphere may be due to a higher fraction (or higher efficiency) of feldspar minerals in the southern dust particles. Some evidence for this has been already found by comparing the immersion freezing efficiency of dust particles from different deserts worldwide (Boose et al., 2016). In these results, the immersion efficiency of dust particles lays mostly between Kaolinite and K-feldspar. The dust samples from sources in the Southern Hemisphere (Australia, Etosha and Atacama milled) have a higher freezing efficiency than most of the samples from the Northern Hemisphere sources including Saharan sources for temperatures below −24°C. Although only four of these samples were studied for higher temperatures, between −23°C and −11°C it was again a sample from the Southern Hemisphere (Atacama milled) which exhibited the highest freezing efficiency. Assuming that the higher freezing efficiency of the southern dust sources may be extrapolated to temperatures above −20°C, the higher immersion efficiency of southern mineral dust, possibly due to higher feldspar fractions, may explain the higher ice occurrence-frequency in the Southern Hemisphere at −15°C. The highly efficient particles, most likely feldspar minerals, would be quickly depleted at temperatures around −15°C and would therefore not interfere with the Kaolinite-Illite(Montmorillonite) differences at −30°C. Furthermore, such a



depletion of highly efficient INP during the transport of dust aerosol may also explain the higher ice occurrence-frequency at the mid-latitudes compared to the high-latitudes for similar mixing-ratios of mineral dust, especially at higher temperatures.
The ageing of dust particles may also reduce the freezing efficiency of dust aerosol during the transport from low to high latitudes. The hypotheses explaining the differences in the freezing behaviour of dust between the Northern and Southern Hemisphere are summarized in Table 2.

## 5.2    Assumptions and uncertainties

In the analysis presented above, certain assumptions were made to assess the potential effect of mineral dust on cloud thermodynamic phase. In this section, these assumptions and the uncertainties that arise from them, as well as the subsequent limitations of the resulting interpretation will be discussed.

Despite the long period (2007-2010) used in the study, a significant fraction of the 5-dimensional space used for our analysis (10 dust deciles, 12 months, 15 temperature bins, 96 latitudes, and 12 longitudes) is sparsely sampled or even contains
missing values. In the high-latitudes, a sampling bias exists towards the respective winter seasons because very few night-time retrievals are available in summer. However, the seasonal variability was not found to be a dominating factor in the day-to-day impact of dust mixing-ratio on the FPR (See S.19 in the Supplement to this article). Furthermore, many factors may contribute to higher standard deviations for the ice occurrence, including:

- Changes in dynamical forcing (e.g., updrafts)
- Temperature changes after cloud glaciation
- Ice sedimentation from above (cloud seeding)
- Cloud vertical distribution within the studied temperature ranges
- Turbulence favouring aerosol mixing and sub-grid temperature fluctuations
- Differences in dust mineral composition and/or size
- Coatings affecting aerosol solubility and freezing efficiency
- Impact of INPs other than dust

Additionally, some issues arise from the coarse spatial resolution used in our study. A high dust mixing-ratio simulated in a gridbox indicated as *cloudy* by the satellite observations does not ensure that the dust is actually mixed with the cloud. The subgrid-distribution of dust relative to the exact cloud position remains unresolved. Higher dust mixing-ratios should be
interpreted as indicator for a higher probability that a significant amount of dust was mixed with a collocated cloud. This mixing may have happened during or before the observation by the satellite. The latter assumes that both the cloud and the dust followed the same trajectory. Overall, at coarse resolutions, the combination of modelled dust concentrations with satellite-retrieved cloud properties cannot guarantee the mixture of aerosol and clouds (R. Li et al., 2017b). Similarly, the atmospheric parameters obtained from the reanalysis may not match the conditions for the exact position of the clouds in the





satellite retrievals. However, the atmospheric parameters are expected to match in average the large-scale conditions influencing the aerosol-cloud interactions.

It cannot be ruled out that the increase in ice cloud occurrence in the Southern Hemisphere for higher dust loading arises from other types of INP such as biogenic aerosol (Burrows et al., 2013; O'Sullivan et al., 2018; Petters and Wright, 2015) or background free-tropospheric aerosol (Lacher et al., 2018), which could be misclassified as mineral dust in the reanalysis.

Similarly, a possible correlation between ice cloud occurrence and the atmospheric conditions leading to the emission and transport of mineral dust should be further investigated (e.g., dusty air masses from land are usually warmer and drier). Another interesting explanation of the results presented in this study could involve the mixing of mineral dust particles with ice nucleation active macromolecules (Augustin-Bauditz et al., 2016). Such particles are in the size of few 10 nm (Fröhlich-Nowoisky et al., 2015) and would therefore not be detected if mixed with dust aerosol.

Meteorological parameters have a larger impact on cloud properties than aerosols do (Gryspeerdt et al., 2016). Different meteorological regimes can change the aerosol-cloud interaction by an order of magnitude. **Fig. 11** shows the mean relative humidity, cloud height and large-scale updraft at −15°C for the different fine-mode dust mixing-ratio deciles and the four latitude bands studied in Sect. 4.4. The correlation between fine-mode dust mixing-ratio from the MACC reanalysis and the RH from the ERA-Interim reanalysis (weighted by cloud volume fraction) was found to be negative (**Fig. 11a**). The decrease

in RH at higher dust mixing-ratios could be associated with drier air masses coming from land. If higher dust mixing-ratios are indeed associated with ice production, then the depletion of water vapor due to the growth of ice particles could also explain the correlation. We expect, however, that the impact of the decreasing RH would rather lead to an underestimation of the role of dust mixing-ratio in cloud glaciation. We note that the RH from the ERA Interim reanalysis represents the conditions at a large-scale and not the conditions at a specific location and the moment of the interaction between dust

aerosol and supercooled cloud droplets. Because of this issue, the role of subgrid RH variability for the results cannot be assessed.

The significant positive correlation found between dust aerosol mixing-ratio and the height of the isotherms (weighted by cloud volume fraction) is also an important source of uncertainty **(Fig. 11b)**. Shifts in the isotherm height, either due to warm dry continental air masses carrying high dust loadings, or the heating effect of absorbing dust aerosol, may indeed

cause a dynamic response of the atmosphere leading to higher isotherms (Gómez-Amo et al., 2014; Meloni et al., 2015; Peris-Ferrús et al., 2017). This could cause clouds to be detected in a higher temperature bin after being glaciated at lower temperatures, thus erroneously suggesting an enhanced glaciation occurrence frequency at higher temperatures. It is crucial for future studies to take into account this possibility when studying the occurrence of ice clouds at a certain isotherm.

It is possible to find cases where the reanalysis and the detected clouds have different temperatures. Nevertheless, a

systematic bias towards lower or higher temperatures would rather have an influence on the average FPR and not on the day-to-day variability of FPR. Furthermore, if absorption of solar radiation by dust aerosol were to cause the observed increase in isotherm height, we would expect a greater correlation in the northern mid-latitudes. The influence of dust-loaded air masses



coming from warmer lower latitudes (where most dust sources are located) offers a more plausible explanation. This could also explain the larger correlation found in the southern high latitudes. More details on the spatiotemporal variability of the
cloud height can be found in the supplement (S12) to this article.

Perhaps the largest difficulty of attributing the observed changes in cloud glaciation to variability in dust concentration is the positive correlation found between dust and the vertical velocity. **Fig. 11c** shows the large-scale vertical velocity from the MACC reanalysis against the fine-mode dust deciles at −15°C. Stronger updrafts are often associated with higher relative humidity. Saturation over liquid water is a necessary condition for the immersion freezing mode, which is believed to be the
most important freezing mode in cloud glaciation, especially at temperatures above −25°C (Westbrook and Illingworth, 2011). Li et al. (2017a) showed that using the monthly values of the CALIPSO-GOCCP product, the FPR at −10, −20 and −30°C is positively (spatially) correlated with the large-scale vertical velocity (at 500 hPa) from the ERA-Interim reanalysis product. The same study showed by using the CALIOP level 2 aerosol layer product that this correlation is similar for different relative aerosol frequencies (dust, polluted dust and smoke combined). However, in the same study, the increase in
FPR for higher vertical velocities (stronger updrafts) was found to be higher for lower relative aerosol frequencies. Furthermore, stronger updrafts favour saturation over liquid water and therefore CCN activation, droplet growth and inhibition of the WBF (Wegener–Bergeron–Findeisen) process. These processes would cause an underestimation of the dust-cloud-phase relationship.

In summary, much of the co-variability between dust, humidity, updrafts, temperature and cloud ice occurrence-frequency is
still poorly understood. However, we expect that the constrains on humidity and static stability minimized most of the biases discussed in this section.

## 6 Conclusions

For the first time, the MACC reanalysis was combined with satellite-retrieved cloud thermodynamic phase to investigate the potential effect of mineral dust as INP on cloud glaciation on a day-to-day basis at a global scale. Satellite products of cloud
thermodynamic phase for the period 2007-2010 were included. We focused on clouds observed at night-time in the mid- and high latitudes. Our main findings can be summarized as follows:

1. Between −36°C and −9°C, day-to-day increases in fine-mode dust mixing-ratio (from lowest to highest decile) were mostly associated with increases in the day-to-day cloud ice occurrence-frequency (FPR) of about 5% to 10% in the mid- and high- latitudes.

2. The response of cloud ice occurrence-frequency to variations in the fine-mode dust mixing-ratio was similar between the mid- and high- latitudes and between Southern and Northern Hemispheres. Moreover, increases in FPR from first to last dust decile were also present in the northern and southern high-latitudes, even though dust aerosol is believed to play a minor role in cloud glaciation in the Antarctic region.



3. Using constraints on atmospheric humidity and static stability we could partly remove the confounding effects due
to meteorological changes associated with dust aerosol.

4. The results also suggest the existence of different sensitivities to mineral dust for different latitude bands. The
north-south differences in ice occurrence-frequency for similar mineral dust mixing-ratios agree with previous
studies on the mineralogical differences between Southern and Northern Hemisphere. A larger fraction of feldspar
in the Southern Hemisphere could explain the differences at −15°C, and the higher freezing efficiency of Illite and
Smectite (more abundant in the Northern Hemisphere) over Kaolinite (more abundant in the Southern Hemisphere)
could explain the differences at −30°C.

We believe these new findings may have an important influence on improving the understanding of heterogeneous freezing
and the indirect radiative impact of aerosol-cloud interactions. The authors hope that the results of this work will also
motivate further research, including field campaigns in remote regions to study the day-to-day variability of cloud
thermodynamic phase and the role of mineral dust in ice formation, satellite-based studies of associated changes in the
radiative fluxes, and modelling studies to test the representation and relevance of specific processes involved in ice
formation and mineral dust transport. Such studies could help to further improve our understanding of the influence of
mineral dust on cloud glaciation and the climate system.

## 7    Author contribution

DV, IT, BH and PS contributed to the design of the study. DV processed the datasets, performed the analysis, designed the
figures and drafted the manuscript. All authors contributed valuable feedback throughout the process. All authors helped
with the discussion of the results and contributed to the final manuscript.

## 8    Competing interests

The authors declare that they have no conflict of interest.

## 9    Acknowledgments

We thank the GOCCP project for providing access to the CALIPSO-GOCCP gridded cloud phase profiles. We thank the
NASA CloudSat project and the CloudSat Data Center for providing access to the 2B-CLDCLASS product. We thank the
ICARE Data and Services Center for providing access to the DARDAR and CloudSat data. We thank the MACC project and
the ERA-Interim science team for providing access to the reanalysis data. All datasets used in the analysis are freely
available at http://climserv.ipsl.polytechnique.fr/cfmip-obs/Calipso_goccp_new.html, http://www.icare.univ-lille1.fr/archive,
http://apps.ecmwf.int/datasets/data/macc-reanalysis/levtype=ml        and        https://apps.ecmwf.int/datasets/data/interim-full-



daily/levtype=pl/ (last access: 13 February 2019). We thank Dr. Albert Ansmann, Dr. Johannes Mülmenstädt and Dr. Julien Delanoë for helpful discussion. The author would like to thank the editor and Anonymous referee #4 for suggesting the inclusion of constraints for humidity and static stability, which greatly improved the accuracy of the results.




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



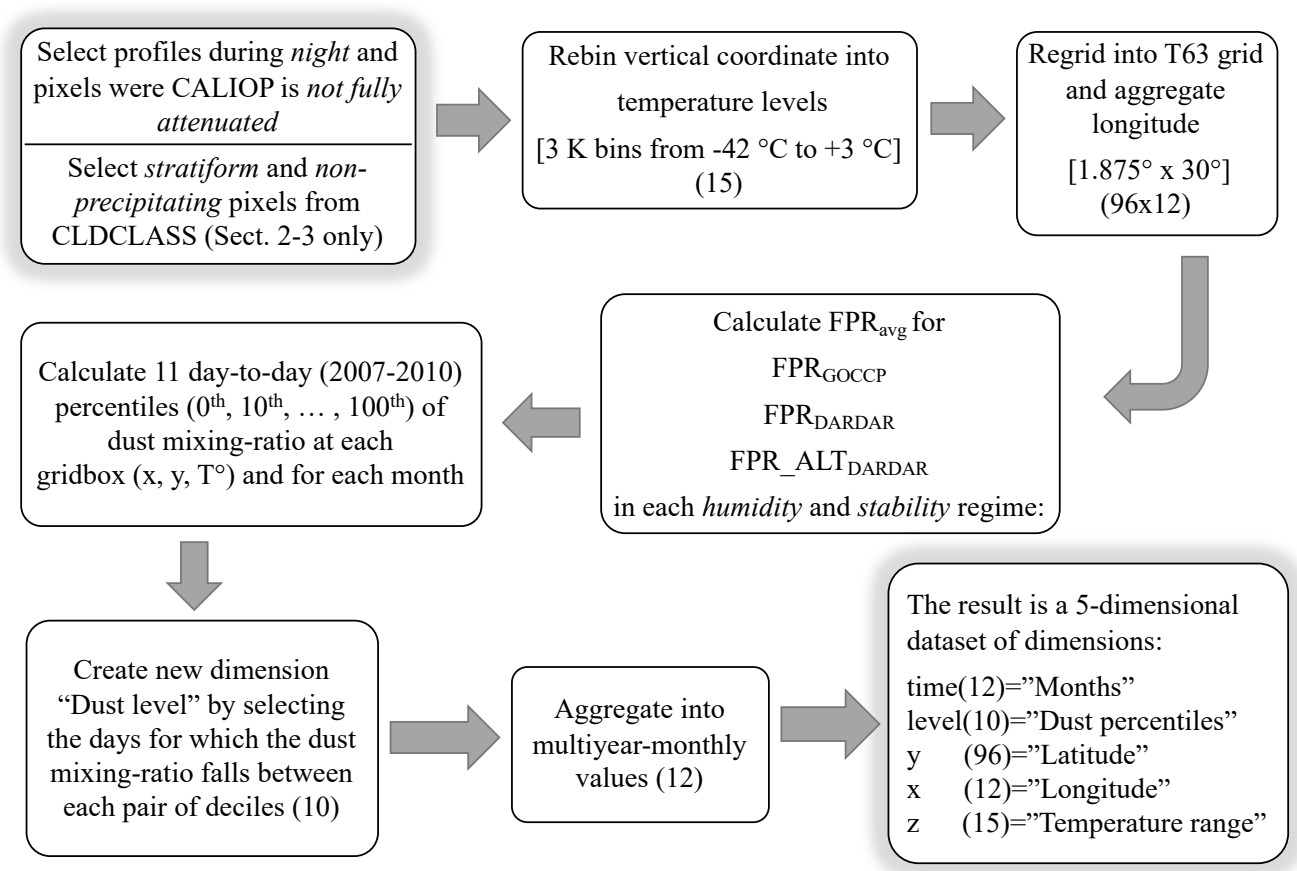


**Figure 1. Flow chart showing the processing steps starting from the raw data (satellite retrievals and model reanalysis) to the dataset used for the analysis.**





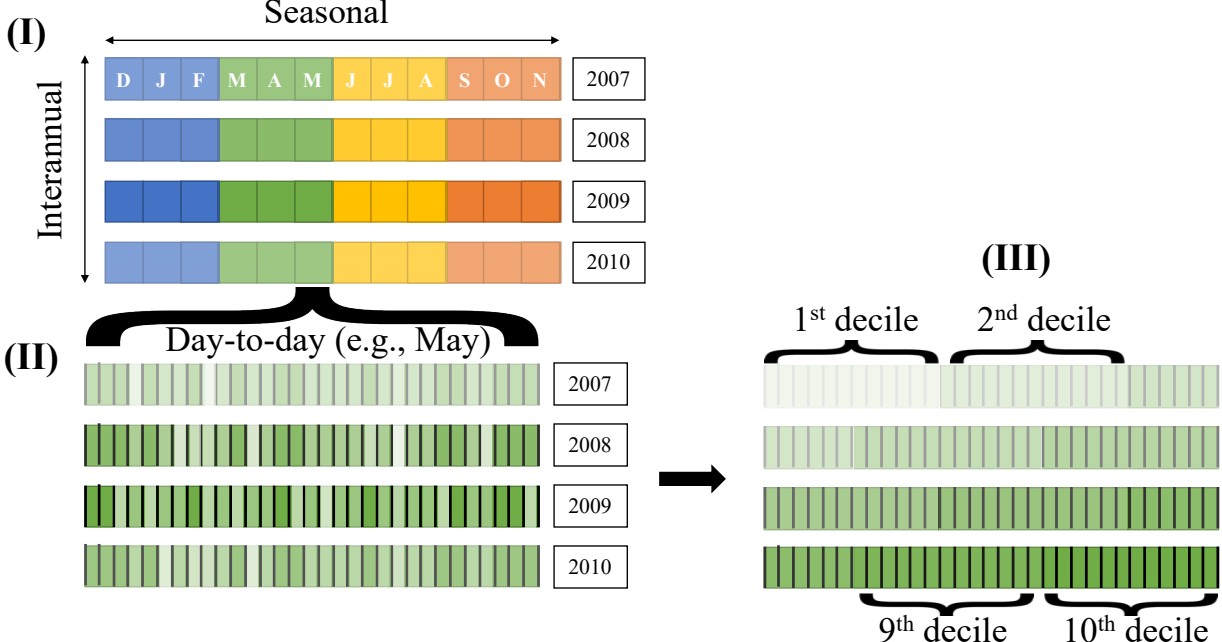

 **Figure 2. Seasonal, day-to-day and day-to-day decile concept as used in this study. For this example, the day-to-day analysis of May contains 124 daily datapoints. In step (I) to (II), only the daily values for one month of the year are selected. In step (II) to (III), these daily values are sorted into 10 different deciles.**






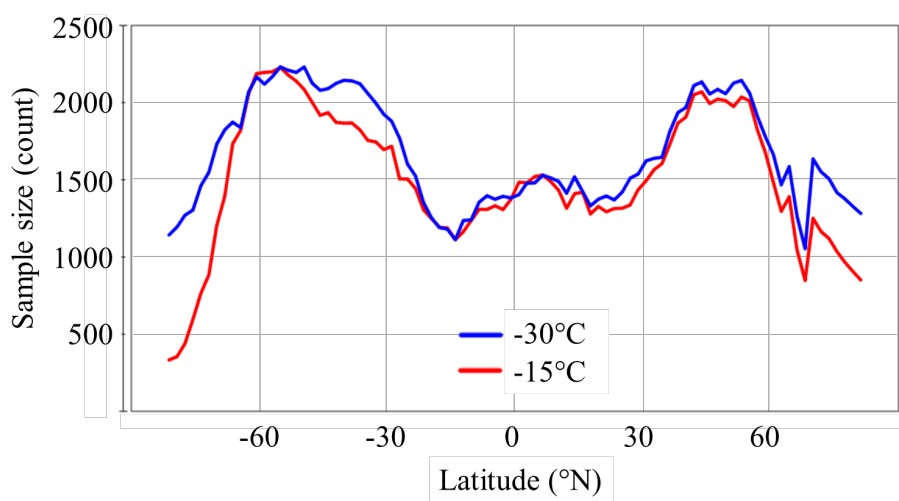

**Figure 3. Sample size of cloud phase (CALIPSO-GOCCP) of each latitude band for −15°C (range −21°C to -9°C) and −30°C (range −36°C to −24°C) for the period 2007-2010. Each count corresponds to a 1.875°×30° gridbox in a 3 K temperature bin at a specific month of the year and inside a specific dust decile. The theoretical maximal sample size for each latitude band is 5760 for a 12 K temperature range.**








**Figure 4. Case study 9:50 UTC Dec 14, 2010. a-b) Cloud volume fraction (GOCCP) for different cloud types (CloudSat cloud classification). c) Fine dust (0.03-0.55 µm) aerosol mixing-ratio (MACC reanalysis), note the logarithmic scale. d-f) Adjusted ice occurrence-frequency derived from the DARDAR-MASK and CALIPSO-GOCCP products. FPR: Frequency phase ratio (ice pixels/total pixels). In FPR_ALT mixed-phase gridboxes (containing both liquid and ice pixels) are reclassified as liquid. White colours represent clear sky. The fields were collocated in a 1.875°x1.875° grid with temperature bins of 3 K each.**






**Figure 5. Global ice cloud occurrence-frequency for stratiform clouds (2007-2010). The average ice occurrence-frequency is weighted by the volume cloud fraction. The fine-mode dust mixing-ratio from the MACC reanalysis corresponds to the range 0.03-0.55 μm and is presented on a logarithmic scale on the right vertical axis. Each temperature bin spans 3 K. The vertical bars show the mean day-to-day standard deviation between different fine-mode dust deciles.**





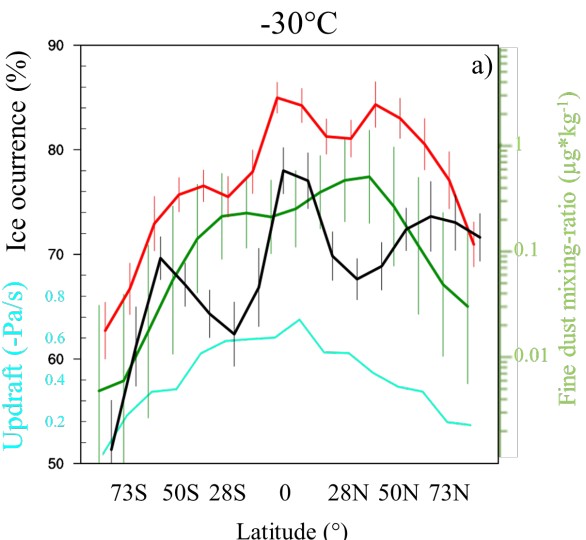

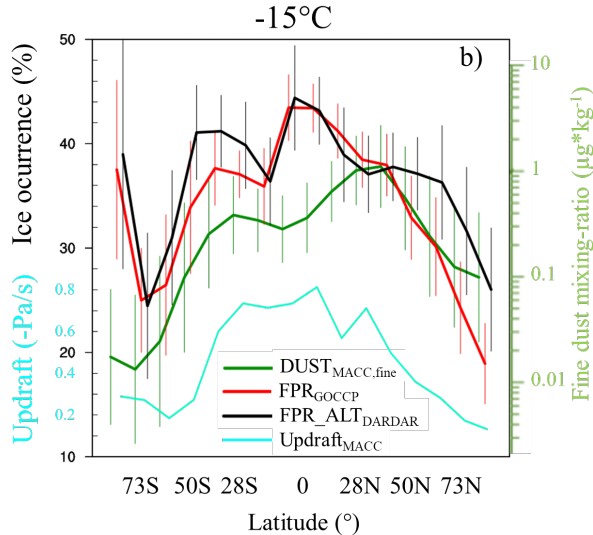

**Figure 6. Zonal mean of stratiform cloud ice occurrence-frequency for (a) −30°C (range −36°C to −24°C) and (b) −15°C (range −21°C to -9°C) averaged over the period 2007-2010. Each datapoint corresponds to a zonal band of 11.25° width. The average fine-mode dust mixing-ratio of each band is also shown on the right vertical axis (note the logarithmic scale). The average large-scale vertical velocity (updraft) from the MACC reanalysis is also shown (cyan axis on the left of each plot). The vertical bars show the mean day-to-day standard deviation between different fine-mode dust deciles. The curves for dust and FPR_ALT$_{DARDAR}$ are slightly shifted left and right, respectively, to fit all vertical bars.**





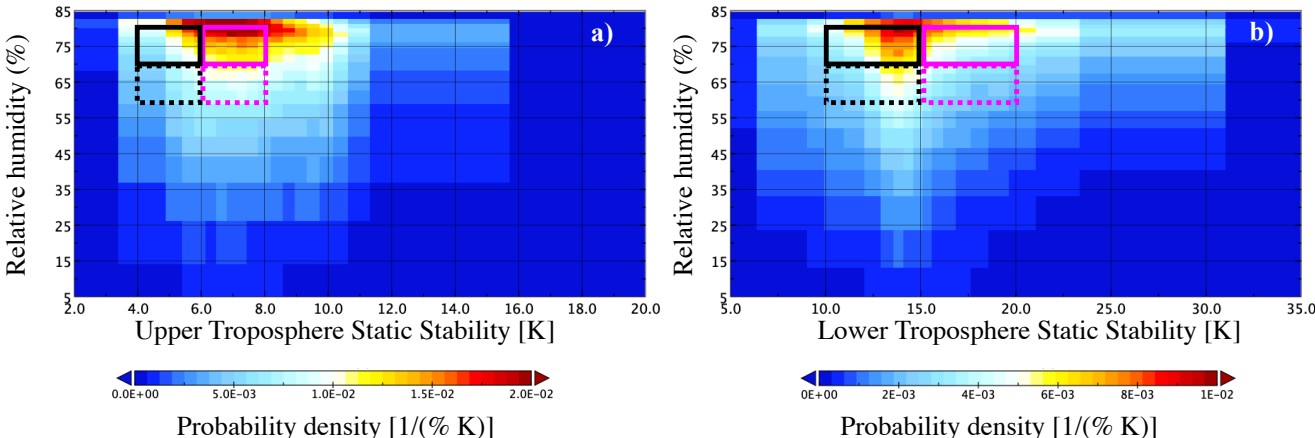

**Figure 7.** Probability histogram at −22 °C (range −27°C to −18°C) for 2007 for different conditions of relative humidity against (a) upper troposphere static stability and (b) lower troposphere static stability. All-sky gridboxes are included for the entire globe. The values for relative humidity are taken from the ECMWF-AUX dataset and the static stability is calculated from the ERA-Interim pressure levels. The magenta and black boxes represent the regimes used in the study.







**Figure 8. Average cloud phase (GOCCP) for the mid-latitude and high-latitude bands averaged between −21°C and -9°C in the period 2007-2010 for different regimes of relative humidity (a,c low RH ; b,c high RH) and lower tropospheric static stability (a,b high LTSS ; c,d low LTSS). The horizontal axis corresponds to the different time deciles (day-to-day variability) of fine-mode dust mixing-ratio (MACC) calculated for each 3 K temperature bin and gridbox (1.875°x30°) and averaged along each 12 K temperature range and latitude band. The vertical bars are positioned at each dust decile and show the mean zonal standard deviation within each latitude band.**

890

895







**Figure 9.** Same as Fig.7 but averaged from −36°C to −24°C and using the upper tropospheric static stability (see Fig. 7a).



**Figure 10. Same as Fig.9 but averaged from −27°C to −18°C.**





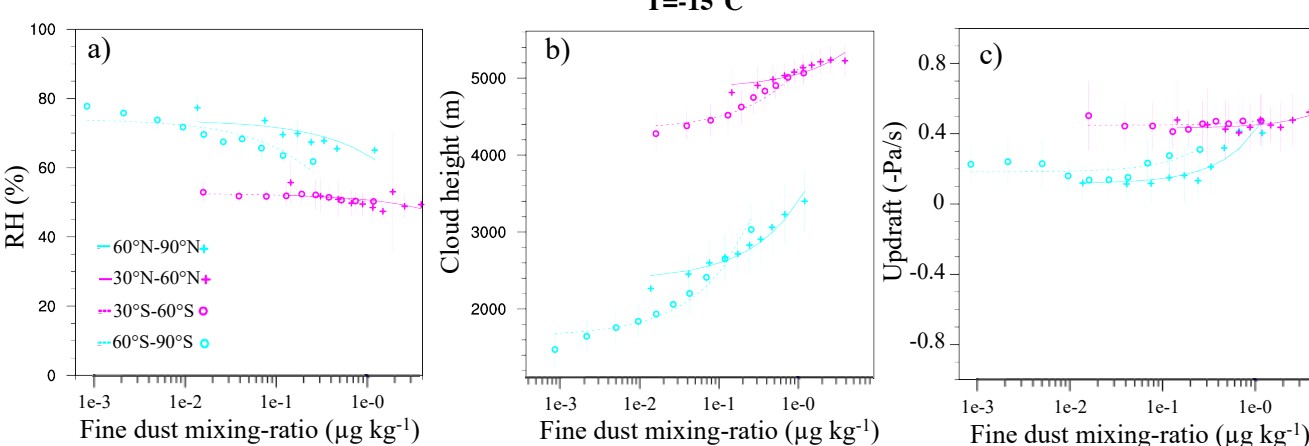

Figure 11. Same as Fig. 8a but for a) ERA Interim relative humidity, b) ECMWF-AUX isotherm height and c) MACC large-scale vertical velocity at −15°C. The average of each variable is weighted by cloud volume.



**Table 1. Summary of the different variables used to assess the Frequency Phase Ratio (FPR).**

| Variable | Ice virgae classified as | Ice and liquid in same gridbox (1.875°) | Ice fraction between 0 and -42°C | Explanation | References |
|---|---|---|---|---|---|
| $FPR_{GOCCP}$ | not detected | ice/liquid (depolarization) | 0 – 100% | Bias towards liquid cloud tops | Cesana and Chepfer (2013) |
| $FPR_{DARDAR}$ | Ice | ice/liquid (radar reflectivity) | 60 – 100% | Ice virgae dominate the cloud phase | Delanoë and Hogan (2008,2010) |
| $FPR\_ALT_{DARDAR}$ | Ice/liquid | liquid | 10 – 90% | Ice virgae are mostly ignored in the gridbox if cloud droplets are also present | - |



920 **Table 2. Summary of the north-south differences in the cloud phase associated with mineral dust observed from the day-to-day statistics for the mid- and high latitudes.**

| Temperature | Range | FPR North-South | Fine dust mixing-ratio | Hypothesis | Related studies |
|---|---|---|---|---|---|
| −15°C | −21 to −9°C (252 to 264 K) | −5 to −3% (high LTSS) | 0.1-1 µg/kg | Higher feldspar fraction (or efficiency) from the sources in the Southern Hemisphere. | Boose et al. (2016), Atkinson et al. (2013) |
| −22°C | −27 to −18°C (246 to 255 K) | ±2% | 0.03-0.3 µg/kg | A transition regime between the immersion freezing of clay minerals and feldspar, or dominance of contact freezing of clay minerals. | Hoose et al. (2008) |
| −30°C | −36 to −24°C (237 to 249 K) | +3 to +5% | 0.03-0.3 µg/kg | Immersion freezing efficiency of Illite/Montmorillonite (Northern Hemisphere) higher than of Kaolinite (Southern Hemisphere). | Hoose et al. (2008), Claquin et al. (2008), Broadley et al. (2012), Murray et al. (2011) |

925