# Peer review of "The day-to-day co-variability between mineral dust and cloud glaciation: A proxy for heterogeneous freezing."

_Atmospheric Chemistry and Physics, 2019_

## Referee Comment (RC1) · Anonymous Referee #2 · 25 Sep 2019

This study is aimed to estimate the role of dust aerosol on the cloud thermodynamic phase using CALIPSO-GOCCP and DARDAR products for cloud phase and the MACC reanalysis for dust mixing ratio. There are some interesting results regarding the relationship between dust and cloud ice. However, I personally found the manuscript to be difficult to follow, which makes it difficult for me to evaluate the scientific merit of this study.

I have the following major concerns for the authors to considered and clarified:

1. Since this paper talks about the role of dust aerosol on the cloud thermodynamic phase, I thought the analyses would focus on mixed-phase clouds. However, mixed-

phase clouds are categorized into liquid clouds in this manuscript (Section 2.3). Does the analysis focus on pure ice clouds? If yes, what about the effect of dust on cloud phase?

2. Why only stratiform clouds are considered in this study?

3. How would the uncertainties in MACC data, such as the significant overestimate of the fine-dust fraction, affect the analysis results?

4. How would the authors ensure the consistency among the different datasets, i.e., satellite products and reanalysis data? And how would the inconsistency affect the analyses and conclusions?

---

## Referee Comment (RC2) · Anonymous Referee #1 · 27 Sep 2019

The manuscript by Villanueva et al. aims at contributing to the interpretation of the role of mineral dust in the formation of ice clouds. In particular, the authors want to demonstrate that the day-to-day che co-variability between mineral dust concentration and cloud glaciation can be used in future as a proxy to better understand heterogeneous freezing mechanism. First of all I must admit that the manuscript is hard to read in several parts, and it takes more than one reading to make sure that the reported content is fully understood. The assessment of the most suitable datasets utilized for the study of heterogeneous freezing as well as the statistics related to the relationship between ice occurrence, updrafts and dust concentration are very interesting. Nevertheless I am not fully sure that this manuscript demonstrates the value of the day-to-day

co-variability as a proxy of the effect on cloud glaciation.

Below I report my general comments.

First of all, the authors have selected data from many difference sources, MACC and ERA interim meteorological reanalysis CALIPSO-GOCCP, DARDAR products from the A-Train satellite constellation. What are the effects on the final results presented in the manuscript of combining these datasource with different speculation (resolutions, sampling, uncertainties, ..)?

The authors often do assumptions and simplifications (eg. use of night time measurements only, neglecting of ice in the mixed phased clouds,...) which can strongly increase the uncertainty of the final results and limit the value of the data interpretation. Cloud phase is mainly regulated by temperature and so it is not clear to me why the authors considered only nighttime measurements. What's the effect of this data selection on the final results?

The scope of the manuscript is to demonstrate that the day-to-day co-variability of dust concentration ice cloud glaciation may be used to quantify the role of dust aerosol on the cloud thermodynamic phase around the globe. I understand that, in order to use the day-to-day variability, the authors removed the seasonal component subtracting the monthly means. I am not sure this is sufficient to remove all the possible variabilities which can affect the selected data. Weather variability for example occurs on a scale of 5-6 days. Can the authors explain how they can assure the data are not affected by any other relevant variability cycle? In addition, it is not clear to me from the reported the description whether the 3K binning can smooth the day-to-day variability (though the binding was needed with respect to the considered dataset) or anyhow mix different observation scenarios, i.e. high dust content and low content but at the same temperature. I think the description in section 3.2 must be clearer.

Many time time the authors state that the presented results can be affected by assumptions or effect not considered but also that several properties, which at regional scale

may have significant difference, should be reconciled by the fact that the static are calculated at the global scale. This is not true for all the variables, RH is an example. There might be strong variations of RH at regional scale in one hemisphere only, which can affect the value of the interpretation of results.

Sampling uncertainties are mentioned a few times by the authors themselves, though these are never quantified. For example, if the regions where the data sample is larger and more complete is resampled to reduce the amount and obtain a dataset of homogenous size across the zonal regions what the effect would be?

Below I report some detailed comments sometimes still of general breath.

Line 177-180: Re-gridding operated here generates also a degradation of the horizontal resolution whose effect on the provided analysis is not quantified. It is not clear to me if this is a real advantage or not.

Line 185: replace "in the study" with the "in the study by Huang et al"

Line2 190-192: please clarify that the choice to ignore ice in mixed phase clouds is an advantage according to the approach your are adopting but also that the authors cannot be sure this has not an impact on the final results.

Lines 264-266: the authors limit their investigation to the altocumulus clouds: can they quantify the impact of this choice on the final results?

Line 322: I think this simplification can create confusion only.

Lines 323-325: though concentration of dust is lower at high altitudes, this does not necessarily indicates that this is due to lower temperatures; this sentence create confusion and solve in a few words a more complicated issue which involves also many other factors, such as atmospheric dynamics and radiative budget. For example, there might be a feedback mechanism influencing the top altitude of aerosols. I think the sentence must be rephrased or otherwise removed.

[Figure]

Line 330: the authors should clarify the reason for the supposed correlation between the maxima observed in the NH and at the tropics, and how the transport of ice clouds downward may occur. Can this be related to any wave activity at the synoptic scale?

Lines 342-343: I do not see the steep increase at the Northern Pole. I ask the authors to clarify.

Lines 345-347: I think the authors may remove this lines, too conjectural; digressions are not needed in that part of the manuscript.

Lines 348-350: how much does the number of data influence you conclusions at the South Pole? The authors should discuss this aspect in the paper.

Lines 352 - 355: the correlation mentioned here between the updraft and the FPR looks not so strong, can the authors provide numbers (i.e. regression coefficient or any other statistical tests)? Given that a correlation with two different parameters of the FPR is studied, is it the case to carry out a partial correlation analysis?

Lines 423 - 427: I agree with the statement provided by the authors though they should acknowledge that in the SH with low USST and high RH the positive correlation is much lower than in other conditions. Can the authors comment a bit more on this aspect? Do the authors envisage a larger contribution in the SH of the homogenous nucleation than in other regions?

Lines 435-439: these lines are to speculative, I'd honestly remove them.

Lines 441-443: Can this results be due to the purer nature of the dust in the SH compared to the NH, where it is often mixed to other aerosol types? In the discussion following to these lines, the authors mention the aged aerosol but never the effect of the aerosol mixing.

Line 502: among the significant number of factors contributing the uncertainty affecting the presented analysis I'd add the limitation to consider only a specific type of cloud type, and only night time observations, as well as the effect of the electric charge of

mineral dust particles.

Line 535 and following: in this section there are few sentences which are very speculative and though these are able put on the table the pletora of different interpretations to te presented data, at the same time, may be not always helpful to the users, also considering that this is not a research article. I suggest to shorten it or arrange in clearer way.

Line 547: something missing in this sentence.

––––––––––––––––––––––

---

## Author Comment (AC1) · 16 Nov 2019

*For clarity, referees' comments are written in black, our comments to each concern are written in blue and extractions from the original paper are written in green.*

**Anonymous Referee #2**

This study is aimed to estimate the role of dust aerosol on the cloud thermodynamic phase using CALIPSO-GOCCP and DARDAR products for cloud phase and the MACC reanalysis for dust mixing ratio.There are some interesting results regarding the relationship between dust and cloud ice. However, I personally found the manuscript to be difficult to follow, which makes it difficult for me to evaluate the scientific merit of this study.

We thank Referee 2 for the useful comments. We believe that the revised version of the manuscript is now easier to follow.

I have the following major concerns for the authors to considered and clarified:

1. Since this paper talks about the role of dust aerosol on the cloud thermodynamic phase, I thought the analyses would focus on mixed-phase clouds. However, mixed-phase clouds are categorized into liquid clouds in this manuscript (Section 2.3).
   o   Does the analysis focus on pure ice clouds?
      ▪   If yes, what about the effect of dust on cloud phase?

Line 131: —" In this case, the pixel is categorized as supercooled or mixed-phase depending on the radar signal, which is assumed a priori to indicate the presence of ice particles. Otherwise, the pixel is categorized as ice (Delanoë et al., 2013; Mioche et al., 2014). For reasons that will become clear later, we will coerce the mixed-phase category into the liquid category."—

This may not have been clear enough in the text. We have now emphasized that we do not mean mixed-phase *clouds* but mixed-phase *pixels*. Coercing mixed-phase pixels (Supercooled droplets and ice particles) into supercooled liquid is a common simplification even when studying mixed-phase clouds — in which the pixels at cloud top are classified as liquid, while the pixels below are classified as ice. In other words, the analysis does include mixed-phase clouds, although they are only detected as such in the DARDAR product.

Nevertheless, the analysis is rather focused on the occurrence ratio between ice and supercooled liquid clouds in the CALIOP-GOCCP product at different temperatures and dust conditions. In this product, mixed-phase clouds are mostly detected as supercooled liquid cloud tops.

2. Why only stratiform clouds are considered in this study?

Firstly, including convective clouds from the study does not change the results significantly (not shown). This is because the phase ratio is calculated as an area ratio between ice and liquid pixels at each temperature. Therefore, the phase ratio is strongly dominated by the phase of stratiform clouds.
Moreover, stratiform clouds are a perfect target to study cloud glaciation thanks to the simpler microphysics compared to convective clouds. Because stratiform clouds are thinner than convective clouds, a larger fraction of the vertical structure can be penetrated by CALIOP.
We have now extended this explanation in the methodology section.

+(new Lines 282-287): *"These cloud types are frequently thin enough to be penetrated by lidar and radar systems and are therefore a good target to study cloud glaciation processes (Bühl et al., 2016; D.Zhang et al., 2010b). Moreover, stratiform clouds have also simpler microphysics compared to convective clouds, where the dynamical forcing is usually stronger."*

3. How would the uncertainties in MACC data, such as the significant overestimate of the fine-dust fraction, affect the analysis results?

We are aware that a significant overestimation exists on the fine-to-coarse dust ratio in the MACC reanalysis (lines 156-157). However, we focused on the relative variations on the dust loading. Therefore, we expect that such overestimations would cause merely a shift in the x-axis with respect to the true values (Fig 8-11), assuming that this overestimation is homogeneous along the dust loading spectrum, which of course would cause some uncertainty. We have now included this issue in the discussion section:

+(new Lines 579-583*):"Furthermore, biases such as the overestimation of the fine-mode dust aerosol in the MACC reanalysis (Ansmann et al., 2017; Kok, 2011) may shift the mixing-ratios shown in Sect. 4.4. However, as long as such biases are not limited to certain meteorologicalconditions, the cloud phase averaged inside each dust decile should remain unaffected."*

4. How would the authors ensure the consistency among the different datasets, i.e., satellite products and reanalysis data?

As discussed in the text, we expect that the assimilation of the total AOD from MODIS in the MACC reanalysis produce a fair estimation of the large-scale aerosol conditions. At least for the Northern Hemisphere, this has been validated with in situ measurements.
As for the different reanalyses, both the ERA-Interim and the MACC reanalysis are based on the IFS model and use a similar assimilation algorithm.
Among the different satellite products, both rely on CALIOP to determine the presence of clouds. Nevertheless, we are aware that several uncertainties remain, for example, between the meteorology in the reanalysis and in the real atmosphere, a topic that is also discussed in the manuscript.

o    And how would the inconsistency affect the analyses and conclusions?
        In the worst-case that the reanalysis is entirely inconsistent with the retrievals of cloud phase, we expect the result would be the lack of correlation between dust and the ice occurrence (Fig 8-10). In other words, given the large dataset included in the study, we expect that mismatches between reanalysis and cloud retrievals would cause an underestimation of the dust-cloud-phase correlation.

We have included these points in the discussion section:

+(new Lines 560-570): "*In general, we expect that the assimilation of the total AOD from MODIS in the MACC reanalysis produce a fair estimation of the large-scale aerosol conditions on a day-to-day basis. At least for the Northern Hemisphere, this has been already validated with in situ measurements (Cuevas et al., 2015). As for the consistency among the different reanalyses, both the ERA-Interim and the MACC reanalysis are based on the IFS model and use a similar assimilation algorithm. Among the different satellite products, both CALIPSO-GOCCP and DARDAR-MASK rely on CALIOP to determine the presence of clouds. Nevertheless, the reader should be aware that several uncertainties remain, for example, between the meteorology in the reanalysis and in the real atmosphere, particularly on the sub-grid scale. In the worst-case that the reanalyses are entirely inconsistent with the retrievals of cloud phase, we expect the result would be the lack of correlation between dust and the ice occurrence (Fig 8-10). In other words, given the large dataset included in the study, we expect that mismatches between reanalysis and cloud retrievals would cause an underestimation — and not an overestimation — of the dust-cloud-phase correlation.*"

---

## Author Comment (AC2) · 16 Nov 2019

*For clarity, referees' comments are written in black, our comments to each concern are written in blue and extractions from the original paper are written in green.*

**Anonymous Referee #1**

The manuscript by Villanueva et al. aims at contributing to the interpretation of the role of mineral dust in the formation of ice clouds. In particular, the authors want to demonstrate that the day-to-day the co-variability between mineral dust concentration and cloud glaciation can be used in future as a proxy to better understand heterogeneous freezing mechanism.

First of all I must admit that the manuscript is hard to read in several parts, and it takes more than one reading to make sure that the reported content is fully understood. The assessment of the most suitable datasets utilized for the study of heterogeneous freezing as well as the statistics related to the relationship between ice occurrence, updrafts and dust concentration are very interesting. Nevertheless I am not fully sure that this manuscript demonstrates the value of the day-to-day co-variability as a proxy of the effect on cloud glaciation.

We thank Referee 1 for thoroughly reading the manuscript and for his/her detailed comments. We believe that the manuscript is now easier to read, after the changes we made in response to the Referee's comments. We also thank the reviewer for his/her encouraging comments about the statistics related to the ice occurrence and dust concentrations, which are indeed the main new idea that we want to present in the manuscript. We regret that the referee is still not convinced of the day-to-day statistical approach, but we hope that our response may satisfy the reviewer's concerns.

Below I report my general comments.

- First of all, the authors have selected data from many difference sources, MACC and ERA interim meteorological reanalysis CALIPSO-GOCCP, DARDAR products from the A-Train satellite constellation. What are the effects on the final results presented in the manuscript of combining these datasource with different speculation (resolutions, sampling, uncertainties, ..)?

> Concerning the different product resolutions, the choice of 3 K bins is based mainly on the resolution of the CPR (480 m) — used in the DARDAR product — and the original 3 K bins of the CALIPSO-GOCCP product.
> In our methodology, a higher resolution (e.g., 1 K bins) would result in a more heterogeneous distribution of the sample size — for any given temperature and day, a larger fraction of gridboxes would contain no clouds — and therefore the statistical uncertainty in the day-to-day statistics would increase. Although the instantaneous satellite products could be resampled to 1 K bins, for the reanalysis the vertical levels would need to be interpolated. Furthermore, some additional technical disadvantages would be associated with a higher resolution. In general, increasing the resolution of the dataset combination would require a new methodology including interpolation of some products. To test if smaller bins could increase the confidence in the results by better constraining the temperature effect on cloud phase, we have performed a quick sensitivity study using the whole range 2007-2010 of the DARDAR-MASK product. In this quick study, we used 3 K bins and 1 K bins to quantify how the vertical resolution affects the variance of the ice-to-liquid ratio. This sensitivity study shows that at -30°C the temporal standard deviation of the FPR decreases by less than 10% in the 1 K bins, despite a decrease of almost 50% in the standard deviation of the temperature inside the temperature bin. Although this represents only a small subset of the data, it shows that a higher vertical resolution not necessarily leads to a decrease in the uncertainty.

> Table R1. Temporal standard deviation (STD) of the ice to liquid ratio from the DARDAR-MASK product by different resolutions (RES) for the temperature bins. The daily values for the whole period 2007-2010 were used. Only global means are shown (96 lat×12 lon)

| INTERVAL | RES | STD | VARIABLE |
|---|---|---|---|
| -30°C TO -33°C | 3K | 0.17 | FPR_DARDAR |
| -30°C TO -31°C | 1K | 0.15 | FPR_DARDAR |
| -31°C TO -32°C | 1K | 0.16 | FPR_DARDAR |
| -30°C TO -33°C | 3K | 0.047 | Temperature |
| -30°C TO -31°C | 1K | 0.028 | Temperature |
| -31°C TO -32°C | 1K | 0.028 | Temperature |

Analogously, using a coarser vertical resolution (e.g., 6 K bins) would allow larger temperature variations. For example, a decrease of 3 K is roughly equivalent to a fivefold increase in INP concentrations (e.g., Niemand et al., 2012). However, because the typical range of the day-to-day variations of dust mixing-ratio is also large — about 1 order of magnitude — we expect that the variability of dust loading should dominate over temperature variations, given a constraint of at least 3 K.

The statistical distribution of the ice ratio also limits the resolution options. Single pixels values of cloud phase from the DARDAR and GOCCP products are binary (1 or 0). Therefore, a minimal sample size is required for the averaged cloud phase —within a certain temperature range, gridbox and percentile of dust— to achieve a normal distribution, which allows interpreting the correlation with dust loading directly.

We have now included this matter in the discussion section (without the table):

+(new Lines 517-528) "*Concerning the vertical resolutions of the different products, the choice of 3 K bins is based on the resolution of the CPR (480 m) — used in the DARDAR-MASK product — and the original 3 K bins of the CALIPSO-GOCCP product. Using a coarser vertical resolution (e.g., 6 K bins) would hinder the assessment of the role of dust as INP. For example, a decrease of 3 K in temperature is roughly equivalent to a fivefold increase in INP concentrations (e.g., Niemand et al., 2012). Because at the mid- and high-latitudes the typical standard deviation of the day-to-day dust mixing-ratio corresponds to roughly a fourfold increase from the mean (See supplement figure S.5), we expect that the variability of dust loading should dominate over temperature variations, given a temperature constraint of 3 K or less. The statistical distribution of the phase ratio also limits the resolution options. The cloud phase values for single pixels in the DARDAR and GOCCP products are binary (1 or 0). Therefore, a minimal sample size is required for the averaged cloud phase ratio — within a certain temperature range, gridbox and percentile of dust — to achieve a normal distribution along time and space, which allows interpreting the correlation with dust loading directly. For this reason, temperature bins smaller than 3 K result in a less normally-distributed cloud phase ratio.*"

- The authors often do assumptions and simplifications (e.g., use of night time measurements only, neglecting of ice in the mixed phased clouds,. . .) which can strongly increase the uncertainty of the final results and limit the value of the data interpretation. Cloud phase is mainly regulated by temperature and so it is not clear to me why the authors considered only nighttime measurements. What's the effect of this data selection on the final results?

As mentioned in section 3.1, we exclude the daytime retrievals from CALIOP to avoid the influence of noise from sunlight scattering on the retrievals (Li et al., 2017). This is a known issue present in the daytime CALIOP retrievals and this data selection is a necessary step to prepare a consistent dataset (e.g., due to sunlight scattering day and night-time CALIOP retrievals are not directly comparable for our purposes). Nevertheless, the first experiments with the original data showed that the inclusion of daytime retrievals lead to lower FPR-dust correlations (not shown).

As for the neglection of ice in mixed-phase clouds, this was only done in the FPR_DARDAR_ALT variable to study the differences between the CALIPSO-GOCCP and DARDAR products. The final results using the GOCCP product are not affected by this simplification.

- The scope of the manuscript is to demonstrate that the day-to-day co-variability of dust concentration ice cloud glaciation may be used to quantify the role of dust aerosol on the cloud thermodynamic phase around the globe. I understand that, in order to use the day-to-day variability, the authors removed the seasonal component subtracting the monthly means. I am not sure this is sufficient to remove all the possible variabilities which can affect the selected data. Weather variability for example occurs on a scale of 5-6 days. Can the authors explain how they can assure the data are not affected by any other relevant variability cycle?

  As correctly pointed out by the reviewer, the day-to-day variability may still include the effects of weather variability. To disentangle the weather variability from the day-to-day variability is indeed an interesting idea. Nevertheless, the number of data samples available is not enough to additionally subclassify the retrievals according to local weather states and, at the same time, to also assess the effect of day-to-day dust variability.

  We are convinced that it is still possible to study the relationship between dust and cloud phase even without excluding cycles shorter than one month. Although cloud phase may be affected by such cycles (e.g., more liquid clouds at convective fronts and more cirrus clouds at the detrainment regions), it is still possible to distinguish between dusty and non-dusty conditions at each point of the weather cycle. Therefore, once we average over the weather cycle — using monthly means inside each dust percentile — it is still possible to observe the dust-cloud-phase relationship.

  These points have been added to the discussion section:

+(new Lines 529-535): "*As mentioned in section 3.5, we excluded the seasonal component of the dust-cloud-phase correlation by calculating the deciles independently for each month of the year. However, shorter cycles (e.g., weather variability) may still have an influence in the variabiliry of dust and cloud phase. Although the cloud phase may be affected by such cycles (e.g., more liquid clouds at convective fronts and more cirrus clouds at the detrainment regions), it is still possible to distinguish between dusty and non-dusty conditions at each point of the weather cycle. Therefore, once we average over the weather cycle — using monthly means inside each dust percentile — we expect the dust-cloud-phase relationship to be dominated by the microphysical effect of dust on cloud phase.*"

- In addition, it is not clear to me from the reported description whether the 3K binning can smooth the day-to-day variability (though the binning was needed with respect to the considered dataset)

  The 3 K can be understood as averaging the satellite pixels between the height of two different isotherms. These isotherms change on a day-to-day basis as estimated by the reanalysis. Therefore, even small day-to-day changes in temperature (e.g., 1 K) are traduced in slight changes in the isotherm heights and in the interval for which the 3 K bins are calculated. Therefore, we do not expect the effect of day-to-day changes in temperature to be an important source of uncertainty in our analysis.

  o … or anyhow mix different observation scenarios, i.e. high dust content and low content but at the same temperature.

  This is an interesting concern. However, the 3K binning is unlikely to produce a significant mixing between different dust scenarios. Even though aerosol plumes can absorb short and longwave radiation, local differences are usually less than 3 K. Even Saharan dust layers has been found to produce temperature differences of only about +2 K in the North Atlantic (Wang and Liu, 2014).
  Although cases of high dust concentration over land could produce higher temperature differences — and therefore a temperature inversion —, we do not expect such rare cases to introduce a significant systematic bias in the results. Neither do we expect temperature inversions to play a significant role at temperatures below -15°C , except perhaps near the poles.

      o    I think the description in section 3.2 must be clearer.

*—"3.2 Regridding and rebinning: Temperature levels and 1.875°×30° gridboxes*

*It will become clear in Sect. 4.2. That the cloud thermodynamic phase is mainly a function of temperature. Anticipating this, temperature bins of 3 K each were used as vertical coordinate throughout the study. The temperature profiles were obtained from the ECMWF-AUX reanalysis for the DARDAR and CLDCLASS products and from the MERRA reanalysis for the GOCCP product.*
*To fill the horizontal gaps between the satellite orbits, we regridded the dataset into a Gaussian T63 grid, aggregating 16 gridboxes along the longitude (1.875°×30°; lat×lon). The Gaussian T63 grid is commonly used in Global Climate Models (Randall et al., 2007) and facilitates future comparisons with global simulations of cloud thermodynamic phase. In section 4.4 and onwards, latitude bands of 30° are used to allow a direct comparison with previous studies (Zhang et al., 2018)."*

We have changed the description to:

+(new Lines 178-190) *"3.2 Regridding and rebinning: Temperature levels and 1.875°×30° gridboxes*

*It will become clear in Sect. 4.2. that the cloud thermodynamic phase is mainly a function of temperature. Anticipating this, temperature bins of 3 K each were used as a vertical coordinate throughout the study. The temperature profiles were obtained from the ECMWF-AUX reanalysis for the DARDAR and CLDCLASS products and from the MERRA reanalysis for the GOCCP product. Thus, the same temperature information was used for each product algorithm and for its postprocessing. The temperature at each profile pixel is interpolated using the information from the reanalyses and between -42°C and +3°C, the pixels are averaged into 3 K intervals.*
*For each of these 3 K intervals, the pixels are then averaged horizontally in a process known as regridding. We regridded the dataset into a Gaussian T63 grid, aggregating 16 gridboxes along the longitude (1.875°×30°; lat×lon) to better fill the horizontal gaps between the satellite orbits. The Gaussian T63 grid is commonly used in Global Climate Models (Randall et al., 2007) and facilitates comparisons with global simulations of the cloud thermodynamic phase. In section 4.4 and onwards, latitude bands of 30° are used to allow a direct comparison with previous studies (Zhang et al., 2018)."*

- Many time the authors state that the presented results can be affected by assumptions or effects not considered but also that several properties, which at regional scale may have significant difference, should be reconciled by the fact that the static stability is calculated at the global scale. This is not true for all the variables, RH is an example. There might be strong variations of RH at regional scale in one hemisphere only, which can affect the value of the interpretation of results. Sampling uncertainties are mentioned a few times by the authors themselves, though these are never quantified.

      The constraints on statistic stability and RH were intended to constrain the temporal and not the regional variability. Therefore, intervals used as constraints are based on the day-to-day meteorological variations and do not consider the regional variability. We agree with the referee that the uncertainty from the regional variability may be a relevant source of uncertainty. However, such regional variations are at least partially accounted by the standard deviation (zonal) shown in Figures 8-11 (error bars). The focus of the study is not the local regional differences but rather the potential hemispheric differences. Therefore, a deeper investigation of smaller-scale regional correlations is out of the scope of this study.

      o    For example, if the regions where the data sample is larger and more complete is resampled to reduce the amount and obtain a dataset of homogenous size across the zonal regions what the effect would be?

      The suggestion is to change the gridbox size ("resample") at each latitude to obtain the same number of data point inside each gridbox. The result would be a highly irregular grid, however, as the reviewer suggests, the data distribution should be much more homogeneous between such gridboxes. Indeed, there would

be some statistical advantages of this approach (e.g., the zonal averaging would be associated with a lower variance). Nevertheless, new problems would arise, as for example a bias towards regions with higher cloud cover — and therefore a larger sample size. Moreover, our analysis is based on the assumption of a regular grid and we consider the implementation of such a challenging dynamic grid structure out of the scope of our study.

Below I report some detailed comments sometimes still of general breath.

- Line 177-180: Re-gridding operated here generates also a degradation of the horizontal resolution whose effect on the provided analysis is not quantified. It is not clear to me if this is a real advantage or not.

—(Lines 183-186) *"To fill the horizontal gaps between the satellite orbits, we regridded the dataset into a Gaussian T63 grid, aggregating 16 gridboxes along the longitude (1.875°×30°; lat×lon). The Gaussian T63 grid is commonly used in Global Climate Models (Randall et al., 2007) and facilitates future comparisons with global simulations of cloud thermodynamic phase. In section 4.4 and onwards, latitude bands of 30° are used to allow a direct comparison with previous studies (Zhang et al., 2018). "*

- As stated in the text, the purpose of this regridding is to ensure enough datapoints in each gridbox. It is correct that this degrades the horizontal resolution. However, the analysis is focused on the latitudinal differences — like the North-South hemispheric contrast — rather than on regional differences.

  In general, due to the explorative nature of the approach, we did not pursue a detailed sensitivity study of the dataset configuration (e.g., temporal, vertical and horizontal resolution). However, during the initial screening, higher horizontal resolutions (e.g. 2x2°) resulted in a too-large zonal variance, due to the binary-nature of the ice-to-liquid ratio. In other words, gridboxes with only few retrievals per day were strongly biased towards single clouds —liquid(0) or ice(1)—, so that the zonal standard deviation was higher than the changes due to dust loading variability.

- Line 185: replace "in the study" with the "in the study by Huang et al"
  - Replaced (Line 194)

- Line2 190-192: please clarify that the choice to ignore ice in mixed phase clouds is an advantage according to the approach you are adopting but also that the authors cannot be sure this has not an impact on the final results.

—(Lines 196–198) *"In this alternative definition, which we will call ALT- DARDAR, only gridboxes (1.875°×30°×3 K) filled with ice pixels are considered as ice (fully glaciated), so that just a single liquid pixel is enough to define a gridbox as liquid (not fully glaciated). One advantage of this marginal definition is that it ignores cloud ice in mixed-phase clouds, which is mostly only detected as such by the DARDAR-MASK product and neglected by the CALIPSO-GOCCP product."*

We have now clarified this statement by adding the following:

+(new Lines 203-204) *"However, this neglection of ice in mixed-phase clouds is only carried out to clarify the differences between the products."*

- Lines 264-266: the authors limit their investigation to the altocumulus clouds: can they quantify the impact of this choice on the final results?

—(Lines 261-266) *"Fig. 4a-b show for the same segment the cloud volume cover (CALIPSO-GOCCP) of clouds classified (2B-CLDCLASS) as cirrus or altocumulus (Fig. 4a) and as altostratus or stratocumulus (Fig. 4b). These cloud types are frequently thin enough to be penetrated by lidar and radar systems and are therefore a good target to study cloud glaciation processes (Bühl et al., 2016; D.Zhang et al., 2010b). "*

○ We are a bit confused by this remark. We are afraid we didn't clarify this point enough. We do not limit the analysis to altocumulus only. Furthermore, these lines refer only to the case study. We have added an introductory statement to improve clarity.

+(new Lines 273-280) *"To better understand the differences between the ice-to-liquid ratio retrieved in the DARDAR-MASK and CALIPSO-GOCCP product, this section provides a detailed case study of a stratiform cloud scenario, in which four stratiform cloud types from the CloudSat classification are included — stratocumulus(low-level clouds), altostratus and altocumulus (mid-level clouds), and cirrus (high-level clouds). Although not present in the case study, Nimbostratus are included in the analysis of cloud phase as well and are particularly important in the high latitudes. Stratus clouds are defined for temperatures above 0°C; therefore, they are not relevant for this study. Finally, the horizontal extension of Cumulus and Deep Convection clouds is very low compared to the stratiform clouds and can be therefore ignored in our study, especially outside the tropics (Sassen and Wang, 2008)."*

- Line 322: I think this simplification can create confusion only.

—*"Additionally, in the DARDAR algorithm water can be still classified as ice at +1.5°C due to the melting layer being set to a wet-bulb temperature (Tw) of 0°C. This allows the detection of ice at temperatures slightly above 0°C dry-bulb temperatures (named simply temperature in this work)."*

We understand this concern, which was already pointed out in the quick reports — "Line 315: please use Tw, remove the sentence in brackets and do not adopt the proposed simplification throughout the text". We agree that the role of Tw (wet-bulb) and Td (wet-dry) in the DARDAR-MASK product is somewhat confusing. However, the use of Tw throughout the analysis of the DARDAR product would aggravate the problem. Tw is only used in the classification algorithm of the DARDAR-MASK product, and its only purpose here is to define the height below which ice is not acceptable anymore. In other words, Tw has only an influence in the Td range +3°C to 0°C, where it can extend the temperature interval where ice is allowed in the classification. This temperature bin is only briefly mentioned to explain the high temperature end in Figure. 5 and is not further considered afterwards."

- Lines 323-325: though concentration of dust is lower at high altitudes, this does not necessarily indicates that this is due to lower temperatures; this sentence create confusion and solve in a few words a more complicated issue which involves also many other factors, such as atmospheric dynamics and radiative budget. For example, there might be a feedback mechanism influencing the top altitude of aerosols. I think the sentence must be rephrased or otherwise removed.

— (Lines 323-325)*"Additionally, the average fine-mode dust mixing-ratio is also shown in Fig. 5. At 0°C the mixing-ratio is five times higher than at −42°C (note the logarithmic right y-axis). This reflects the fact that dust mixing-ratios tend to be lower at higher altitudes where temperatures are lower. However, there are important exceptions to this, such as in the long-range transport of dust layers over the ocean."*

*We have rephrased as follows:*

+(new Lines 344-348)*" Additionally, the average fine-mode dust mixing-ratio is also shown in Fig. 5. At the height of the 0°C isotherm, the mixing-ratio is on average higher than at the −42°C isotherm (note the logarithmic right y-axis). This reflects the fact that, on average, dust mixing-ratios tend to be higher near the dust sources at the surface. However, this does not imply any general relationship between dust and temperature. Moreover, instant vertical profiles of dust loading and temperature may differ greatly from this average, especially in the long-range transport of dust plumes."*

- Line 330: the authors should clarify the reason for the supposed correlation between the maxima observed in the NH and at the tropics, and how the transport of ice clouds downward may occur. Can this be related to any wave activity at the synoptic scale?

—*"These maxima are probably associated with the enhanced homogeneous freezing in the tropics at temperatures below −40°C and the resulting downward transport of cloud ice."*

First of all, we must clarify that is not the scope of the paper to clarify the mechanisms of ice production in the tropics. Our assumption is based on previous global climate simulations studies, where the main source of cloud ice below the -42°C isotherm is the ice detrained from convective outflows (Gasparini and Lohmann, 2016). These ice particles may be produced by the rapid injection of cloud droplets to temperatures lower than -42°C. Ice particles tend to grow and sediment faster than cloud droplets, and the associated downdrafts may enhance the downward transport of the detrained cloud ice. Because convection in the mid-latitudes is closely associated with cyclone activities, the synoptic-scale may indeed play an important role in the large-scale correlation between updrafts and cloud ice. We encourage future studies seeking to investigate this possibility.

- Lines 342-343: I do not see the steep increase at the Northern Pole. I ask the authors to clarify.

    We are a bit puzzled by this remark. The steep increase described in the text is refers to the Southern Pole. We have clarified this as following.

—( Lines 348-349)"For both variables, a local minimum near 73°S is followed by a steep increase at 84°S"

This was changed to

+(new Lines 363-365) "Moreover, the FPRGOCCP at −15°C is lower than the FPR_ALTDARDAR at the southern mid-latitudes and northern high-latitudes. In the southern high latitudes, for both variables, a local minimum near 73°S is followed by a steep increase at 84°S."

[Figure]

- Lines 345-347: I think the authors may remove this lines, too conjectural; digressions are not needed in that part of the manuscript.

(Lines 351-353)—"The predominance of ice clouds in Antarctica has been already pointed out earlier in the literature (Ardon-Dryer et al., 2011; Bromwich et al., 2012). Incoming air masses from the ocean may carry higher concentrations of INP like biogenic aerosol (Saxena, 1983), Patagonian soil dust or Australian black carbon (Bromwich et al., 2012). "

    We have removed these lines.

- Lines 348-350: how much does the number of data influence your conclusions at the South Pole? The authors should discuss this aspect in the paper.

(Lines 355-358)— "Similarly, it has been shown that the orographic forcing in Antarctica can lead to high ice water contents for maritime air intrusions (Scott and Lubin, 2016). In other words, maritime air intrusions associated with higher temperatures, higher concentrations of INP and stronger vertical motions could explain the observed pattern in the southern polar regions."

    We recognize that these lines are somewhat speculative. We agree with the referee, that the low sample size near the South Pole (Fig. 3 and supplement material s14.b) together with the low altitude of the -15°C isotherm (s12.b) hinders more robust statistics. For example, at –15°C, the zonal standard deviation of the FPR significantly increases from 60°S towards the South Pole — from about ±0.08 to ±0.16 in Fig.6a — at the same time that the sample size decreases from 2200 to 300 (Fig.3).

    We have now included this issue in the text.

    + (new Lines 370-374) "*However, the low sample size near the South Pole (**Fig. 3** and supplement material **S.14.b**) and the low altitude of the -15°C isotherm (**S.12.b**) result in a lower confidence in the results for this region. For example, at −15°C, the zonal standard deviation of the FPR significantly increases from 60°S towards the South Pole — from about ±0.08 to ±0.16 in Fig.6a — at the same time that the sample size decreases from 2200 to 300 (**Fig.3**).*"

- Lines 352 - 355: the correlation mentioned here between the updraft and the FPR looks not so strong, can the authors provide numbers (i.e. regression coefficient or any other statistical tests)?

—"The pattern of the mean large-scale vertical velocity (MACC reanalysis) of the clouds studied is particularly similar to the FPR at −15°C. Moreover, the spatial correlation between large-scale updraft velocity at 500 hPa is positively correlated to the occurrence-frequency of ice clouds at −20°C (Li et al., 2017a). In other words, both the dust mixing-ratio and the large-scale vertical velocity seem to be positively correlated (spatially) to FPR. There are some plausible explanations for this: "

We have calculated the regression coefficient associated with the zonal averages and the 30°×1.875° gridbox averages and we have also emphasized that we suggest only a large-scale correlation with the average updraft velocity. We have changed these lines as follows:

+(new Lines 375-380)"The time-averaged large-scale vertical velocity (MACC reanalysis) of the clouds studied is regionally correlated with the FPR at −15°C — with a pearson correlation coefficient of 0.47 using zonal averages and of 0.31 using the 30°×1.875° gridbox averages. Moreover, in another study, the spatial correlation between large-scale updraft velocity at 500 hPa was also found to be positively correlated (spatially) to the occurrence-frequency of ice clouds at −20°C (Li et al., 2017a). In other words, both the dust mixing-ratio and the large-scale vertical velocity appear to be to some extent correlated (spatially) to the FPR. There are some plausible explanations for this: "

- Given that a correlation with two different parameters of the FPR is studied, is it the case to carry out a partial correlation analysis?

  The vertical velocity from the reanalysis is only a large-scale estimation and it may not coincide with the instant position of the clouds retrieved from the satellite products. For the same reason, LTSS and RH are better parameters to evaluate the possible influence of convection and constrain the influence of dynamics in the dust-cloud-phase correlation. Within this perspective, when we study the response of cloud-phase to different aerosol concentrations at constant LTSS and RH we fulfil the same objective that of a partial correlation analysis. As for the effect of dynamics (i.e., effect of updraft at a constant aerosol loading), this analysis has been already carried out in previous studies (e.g., Li et al., 2017) and is not the focus of this study.

- Lines 423 - 427: I agree with the statement provided by the authors though they should acknowledge that in the SH with low UTSS and high RH the positive correlation is much lower than in other conditions.

[Figure]

(Lines 423-428)—" For dust mixing ratios between 0.1 and 1.5 µg kg$^{-1}$, the cloud ice occurrence-frequency at −30°C increase by about +5%. The highest increase is found for the northern latitudes. However, the results from the southern mid-latitudes contradict the notion that the INP activity of mineral dust is of secondary importance in the Southern Hemisphere due to low dust aerosol concentrations (Burrows et al., 2013; Kanitz et al., 2011). Nevertheless, recent studies have acknowledged that the importance of mineral dust in the southern latitudes still cannot be ruled out (Vergara-Temprado et al., 2017) "

- Can the authors comment a bit more on this aspect?

We agree with the referee. However, because the difference is only evident for the southern mid-latitudes, it is very difficult to speculate about the reason behind it. Because this difference is not found in the NH, it is difficult to attribute the effect to the stability and humidity conditions. However, the correlation seems to vary little between the regimes, and therefore, it suggests that the positive correlation is consistent for the different cloud-forming conditions.

- o Do the authors envisage a larger contribution in the SH of the homogenous nucleation than in other regions?
  We consider that still much investigation is needed before taking a stand in this question. The relative contribution of homogeneous and heterogeneous freezing — and of the different INP types — is still a matter of debate (Barahona et al., 2017; Dietlicher et al., 2018), especially in the mixed-phase regime (temperature range 0°C to −42°C). Furthermore, even if mineral dust was not a dominant INP in the SH, other particles like marine organic aerosols could still represent important INP and influence cloud ice formation.

- Lines 435-439: these lines are to speculative, I'd honestly remove them.

(Lines 435-439)—"In general, for temperatures between −36°C and −9°C, higher fine-mode dust mixing-ratios are associated with an increasing cloud ice occurrence-frequency. The results suggest that only the lower static stability at −15°C has a strong influence on the relationship between mineral dust and cloud ice. This is may be a consequence of the dynamic component of the atmospheric stability at lower temperatures (e.g., gravity waves), which is not included in the static stability parameter. "

Removed.

- Lines 441-443: Can these results be due to the purer nature of the dust in the SH compared to the NH, where it is often mixed to other aerosol types? In the discussion following to these lines, the authors mention the aged aerosol but never the effect of the aerosol mixing.

(Lines 441-443)—"However, against our expectations, for similar dust loadings the cloud ice occurrence-frequency at −15°C was higher in the Southern than in the Northern Hemisphere."

- o This is a very interesting point. Indeed, the higher concentrations of sulphate in the NH are believed to produce coating in dust aerosol and deactivate its freezing potential. It is therefore not difficult to imagine that mixing with other types of aerosols may cause a similar effect. We do mention in the text the potential role of biogenic aerosol mixed with dust aerosol, which would have an enhancing effect in the freezing potential. When we mentioned aged aerosol we refer mostly to the internal mixing (coating) of sulphate with dust aerosol. We have mention this explicitly.

  +(new Line 509) *"The ageing (e.g., internal mixing with sulfate or "coating") of dust particles may also reduce the freezing efficiency of dust aerosol during the transport from low to high latitudes."*

- Line 502: among the significant number of factors contributing the uncertainty affecting the presented analysis I'd add the limitation to consider only a specific type of cloud type, and only night time observations, as well as the effect of the electric charge of mineral dust particles.

  - o We have added these limitation to the list:
  +(new Lines 542-549)"

    - *Changes in dynamical forcing (e.g., updrafts) and **cloud regimes***
    - *Temperature changes after cloud glaciation (e.g., latent heat release)*
    - *Ice sedimentation from above (cloud seeding), and INPs other than dust*

- *Cloud vertical distribution within the studied temperature ranges*
- *Turbulence favouring aerosol mixing and sub-grid temperature fluctuations*
- *Differences in dust mineral composition, **electric charge** and/or size*
- *Coatings (e.g. Sulfate) affecting aerosol solubility and freezing efficiency*
- ***Subsetting of the data (e.g., only night-time retrievals)"***

- Line 535 and following: in this section there are few sentences which are very speculative and though these are able put on the table the pletora of different interpretations to the presented data, at the same time, may be not always helpful to the users, also considering that this is not a research article. I suggest to shorten it or arrange in clearer way.
  - To improve the clarity, we have shortened these paragraphs as follows:
  -

+(new Lines 583-601)" *In general, meteorological parameters have a larger impact on cloud properties than aerosols do (Gryspeerdt et al., 2016). For example, different updraft regimes can change the aerosol-cloud interactions in warm clouds by an order of magnitude. Therefore, it is important to study how such meteorological parameters relate to the dust aerosol loading. With this purpose, Fig. 11 shows the mean relative humidity, cloud height and large-scale updraft at −15°C for the different fine-mode dust mixing-ratio deciles and for the four latitude bands studied in Sect. 4.4. Firstly, the correlation between fine-mode dust mixing-ratio from the MACC reanalysis and the RH from the ERA-Interim reanalysis — weighted by cloud volume fraction — was found to be negative (Fig. 11a). We note that the RH from the ERA-Interim reanalysis represents the conditions at a large-scale and not the conditions at a specific location and the moment of the interaction between dust aerosol and supercooled cloud droplets. Still, this relationship is consistent with the intuition that dust is mostly associated with drier air masses. Second, The significant positive correlation found between dust aerosol mixing-ratio and the height of the isotherms (weighted by cloud volume fraction) points to an important source of uncertainty (Fig. 11b). This could be due to clouds being detected in a higher temperature bin after being glaciated at lower temperatures, thus erroneously suggesting an enhanced glaciation occurrence frequency at higher temperatures. Therefore, it is crucial for future studies to take into account this possibility when studying the occurrence of ice clouds at a certain isotherm. More details on the spatiotemporal variability of the cloud height can be found in the supplement (S12) to this article. Lastly, Fig. 11c shows a positive correlation between the fine-mode dust and the large-scale vertical velocity from the MACC reanalysis at −15°C. Updrafts favour saturation over liquid water and therefore CCN activation, droplet growth and inhibition of the WBF (Wegener–Bergeron–Findeisen) process. Therefore, a positive dust-updraft correlation could lead to an underestimation of the dust-cloud-phase relationship.*"

Line 547: something missing in this sentence.

—"It is possible to find cases where the reanalysis and the detected have different temperatures. "

Meant was:

+"It is possible to find cases where the reanalysis and the detected **clouds** have different temperatures. "

However, we have removed this line after the previous suggestion to shorten this section.

---

## Referee Report (RR1)

After the first review stage, the manuscript by Villanueva et al. has been improved mainly clarifying in the text a few parts which were unclear and adding a few aspects which were missing and related to sources of uncertainty and limitations within the proposed approach to quantify the impact of mineral dust on the day-to-day variability of stratiform cloud glaciation.

The main manuscript goal is to introduce a new "metric" to quantity the indirect radiative impact of aerosol-cloud interactions.

Therefore, they have the unique opportunity to discuss they method on a rigorous quantitative basis, despite of the intrinsic limitations of the utilized datasets and the contingencies in their interpretation. I am still convinced that the value of using the day-to-day variability to quantify the impact of mineral dust is not fully demonstrated given the number of unquantified factors and uncertainty contributions. Nevertheless, the manuscript is interesting though it must report an analysis which will results incomplete, per its nature, being an innovative idea proposed to the community but based on not "tailored" data. This would not be an issue if all the assessable factors playing a role would be properly quantified. This can guarantee the validity of the results within the uncertainty margins.

It must be stressed that the presented analysis may strongly depend on the utilized dataset, thus loosing of generality, for example because of missing daytime data.

In summary, I could say that I am not fully satisfied by the changes in the updated version of the manuscript. I think some additional work was requested and this was not done. Several aspects touched by the referees have been solved by the authors with the typical statement "we expect that….." without any quantification.

Trying to be concrete in the benefit of the paper and considering the conclusions of the manuscript, I'll try to provide final recommendations of the minimum work which to my opinion must be added in order to provide a convincing message which can really stimulate future studies.

Anyhow, I will not contest the editor final decision if his overall opinion on the review process is satisfying.

1. *Between –36°C and –9°C, day-to-day increases in fine-mode dust mixing-ratio (from lowest to highest decile) were mostly associated with increases in the day-to-day cloud ice occurrence-frequency (FPR) of about 5% to 10% in the mid- and high- latitudes.*

This conclusion relates to the night time data only, this is due to intrinsic dataset limitations (sensor issue and the related period). The authors claim that the presented method can be applied in general but I think they miss in their data the cloud diurnal cycle which is not averaged out by the monthly mean and must be considered in the quantification of the aerosol-cloud-radiation effects.

Nevertheless, to requires the use of more recent CALIPSO data not compromised by spurious effects is too demanding.

For other cycles, possibly present in the data, I acknowledge that, as the authors states, it might be still possible to distinguish between dusty and non-dusty conditions at each point of the weather cycle but the uncertainty affecting their conclusion is not quantified.

*2. The response of cloud ice occurrence-frequency to variations in the fine-mode dust mixing-ratio was similar between the mid- and high- latitudes and between Southern and Northern Hemispheres. Moreover, increases in FPR from first to last dust decile were also present in the northern and southern high-latitudes, even though dust aerosol is believed to play a minor role in cloud glaciation in the Antarctic region.*

The dataset is quite heterogeneous in terms of samples in the different zonal regions and in particular at the South Pole. I suggested a re-gridding of the data in an irregular way which can enlarge the sampling where it is poorer, reducing the related uncertainties. Though this could be not the best method, a new way to make the authors' investigation more robust is needed. Otherwise the presented results are too driven by the dataset limitations. I saw the authors added a paragraph to stress the limitations in the general validity of their results; this can be considered sufficient.

*3. Using constraints on atmospheric humidity and static stability we could partly remove the confounding effects due to meteorological changes associated with dust aerosol.*

This is a point where to my opinion an addition effort is required.

Here, a multivariate analysis (or anything similar) could tell us more and this should be done to give more value to the manuscript. in order to quantify the influence of static stability and humidity on the dust-cloud-phase relationship, a different and organic statistical approach is needed. Same applies to the correlation between dust mixing-ratio and the large-scale vertical velocity, where the authors provide in their answer the calculation or the Pearson's coefficient which reveal a faint correlation.

To support the authors' speculations, often interesting, a broader statistical analysis should be performed to strengthen the final message.

*4. The results also suggest the existence of different sensitivities to mineral dust for different latitude bands. The north-south differences in ice occurrence-frequency for similar mineral dust mixing-ratios agree with previous studies on the mineralogical differences between Southern and Northern Hemisphere. A larger fraction of feldspar in the Southern Hemisphere could explain the differences at −15°C, and the higher freezing efficiency of Illite and Smectite (more abundant in the Northern Hemisphere) over Kaolinite (more abundant in the Southern Hemisphere) could explain the differences at −30°C.*

This is a very interesting speculation and I think, even though it would be valuable, no additional effort is needed for this part of the discussion.

Finally, I ask the authors to more clearly mention in the paper the ongoing debate on the relative contribution of homogeneous and heterogeneous freezing, using also the reference mentioned in their answer (Barahona et al., 2017; Dietlicher et al., 2018).

---

## Author Response (AR2)

After the first review stage, the manuscript by Villanueva et al. has been improved mainly clarifying in the text a few parts which were unclear and adding a few aspects which were missing and related to sources of uncertainty and limitations within the proposed approach to quantify the impact of mineral dust on the day-to-day variability of stratiform cloud glaciation.

The main manuscript goal is to introduce a new "metric" to quantity the indirect radiative impact of aerosol-cloud interactions.
Therefore, they have the unique opportunity to discuss they method on a rigorous quantitative basis, despite of the intrinsic limitations of the utilized datasets and the contingencies in their interpretation. I am still convinced that the value of using the day-to-day variability to quantify the impact of mineral dust is not fully demonstrated given the number of unquantified factors and uncertainty contributions. Nevertheless, the manuscript is interesting though it must report an analysis which will results incomplete, per its nature, being an innovative idea proposed to the community but based on not "tailored" data. This would not be an issue if all the assessable factors playing a role would be properly quantified. This can guarantee the validity of the results within the uncertainty margins.

It must be stressed that the presented analysis may strongly depend on the utilized dataset, thus loosing of generality, for example because of missing daytime data.
In summary, I could say that I am not fully satisfied by the changes in the updated version of the manuscript. I think some additional work was requested and this was not done. Several aspects touched by the referees have been solved by the authors with the typical statement "we expect that....." without any quantification.

Trying to be concrete in the benefit of the paper and considering the conclusions of the manuscript, I'll try to provide final recommendations of the minimum work which to my opinion must be added in order to provide a convincing message which can really stimulate future studies.

Anyhow, I will not contest the editor final decision if his overall opinion on the review process is satisfying.

We regret to hear that the referee is not fully satisfied with the new changes in the manuscript. We have tried to address his/her concerns in detail and hope that our response may convince the referee that we have considered his/her suggestions seriously.

*1. Between −36°C and −9°C, day-to-day increases in fine-mode dust mixing-ratio (from lowest to highest decile) were mostly associated with increases in the day-to-day cloud ice occurrence- frequency (FPR) of about 5% to 10% in the mid- and high- latitudes.*

This conclusion relates to the night time data only, this is due to intrinsic dataset limitations (sensor issue and the related period). The authors claim that the presented method can be applied in general but I think they miss in their data the cloud diurnal cycle which is not averaged out by the monthly mean and must be considered in the quantification of the aerosol-cloud-radiation effects. Nevertheless, to requires the use of more recent CALIPSO data not compromised by spurious effects is too demanding.

For other cycles, possibly present in the data, I acknowledge that, as the authors states, it might be still possible to distinguish between dusty and non-dusty conditions at each point of the weather cycle but the uncertainty affecting their conclusion is not quantified.

As the referee mentions, at the time of this study, the available day-time CALIPSO products were affected by sunlight backscattering. We thank the referee for mentioning the availability of more recent CALIPSO data.

Unfortunately, to accurately quantify the effect of the weather cycle on dust and cloud phase — and therefore in the dust-cloud-phase relationship — one would have to first find a method to correctly determine the stage within the weather cycle at each retrieval, for example using surface pressure as a proxy. This would be undoubtedly valuable. However, we were not able to find a significant correlation between the reanalysis surface pressure and cloud phase. Therefore, we believe a more complex approach should be developed with this purpose. We still believe that such developments fall outside the scope of our study.

Although a proxy for cloud-lifetime may also help in such an approach, choosing a correct parameter for this purpose is challenging (e.g., Witte et. al, 2014; doi:10.5194/acp-14-6729-2014). Arguably, total water content would be the best option to estimate the cloud-lifetime in the retrievals. Then, by comparing clouds with similar cloud lifetime, the artefacts related to the weather cycle could be somewhat constrained. However, there is no warranty that such a constraint would be appropriate. We are enthusiastic that our work could motivate precisely this kind of research.

Heterogeneous freezing itself may drastically shorten the lifetime of clouds. Therefore, the authors see no simple approach to separate the variability associated with the weather cycle from the variability related to dust aerosol. We acknowledge, however, that the former may eventually dominate the latter.

*2. The response of cloud ice occurrence-frequency to variations in the fine-mode dust mixing-ratio was similar between the mid- and high- latitudes and between Southern and Northern Hemispheres. Moreover, increases in FPR from first to last dust decile were also present in the northern and southern high-latitudes, even though dust aerosol is believed to play a minor role in cloud glaciation in the Antarctic region.*

The dataset is quite heterogeneous in terms of samples in the different zonal regions and in particular at the South Pole. I suggested a re-gridding of the data in an irregular way which can enlarge the sampling where it is poorer, reducing the related uncertainties. Though this could be not the best method, a new way to make the authors' investigation more robust is needed. Otherwise the presented results are too driven by the dataset limitations. I saw the authors added a paragraph to stress the limitations in the general validity of their results; this can be considered sufficient.

We share the concerns of the referee and we are glad that the changes in the manuscript could address this issue properly.

*3. Using constraints on atmospheric humidity and static stability we could partly remove the confounding effects due to meteorological changes associated with dust aerosol.*

This is a point where to my opinion an addition effort is required. Here, a multivariate analysis (or anything similar) could tell us more and this should be done to give more value to the manuscript. in order to quantify the influence of static stability and humidity on the dust-cloud-phase relationship, a different and organic statistical approach is needed. Same applies to the correlation between dust mixing-ratio and the large-scale vertical velocity, where the authors provide in their answer the calculation or the Pearson's coefficient which reveal a faint correlation. To support the authors' speculations, often interesting, a broader statistical analysis should be performed to strengthen the final message.

We appreciate the suggestion from the referee and understand why a multivariate analysis would seem appropriate to analyse the intercorrelation between cloud phase, dust loading and meteorology (updraft, RH and stability).

However, some issues hinder the application of such a multivariate analysis:

- Non-normal distribution of dust loading and ice cloud frequency: While updraft, RH, and stability are normal-distributed (temporally), cloud phase follows mostly a binary distribution (0: liquid, 1:ice) and dust loading is strongly skewed and even follows a gamma distribution at the pristine regions in the Southern Hemisphere. Most approaches usually applied in multivariate analysis (e.g., multi-regression, partial derivatives,…) are aimed at normal variables. Applying such methods to our non-normal variables can be dramatically misleading (see for example Hauke et al. 2011; https://doi.org/10.2478/v10117-011-0021-1).

- Limited sample size: This is also related to the non-normal distribution of cloud-phase. We need first to aggregate the data to obtain a normal-distributed cloud-phase variable (similar to a rank correlation). As a result, the remaining sample size is already low — and the statistics noisy — after separating the retrievals between different dust deciles. The constraints on RH and SS were only possible after ensuring that each regime would contain about 10% of the data or more. A narrower regime definition (i.e., closer to a partial derivative) would result in weak correlations, due to the binary-like distribution of cloud-phase, which can only be ignored when the sample sizes are large enough. For the same reasons, a multivariate-rank-correlation would require a lower resolution for the dust conditions (e.g., dust quartiles instead of deciles). This would miss the main focus of the study, which is getting a first glimpse at the correlation between dust aerosol and cloud-phase.

However, we do agree with the referee that a new approach is needed in future studies focusing on the intercorrelation of meteorology, aerosols and cloud phase.

*4. The results also suggest the existence of different sensitivities to mineral dust for different latitude bands. The north-south differences in ice occurrence-frequency for similar mineral dust mixing-ratios agree with previous studies on the mineralogical differences between Southern and Northern Hemisphere. A larger fraction of feldspar in the Southern Hemisphere could explain the differences at −15°C, and the higher freezing efficiency of Illite and Smectite (more abundant in the Northern Hemisphere) over Kaolinite (more abundant in the Southern Hemisphere) could explain the differences at −30°C.*

This is a very interesting speculation and I think, even though it would be valuable, no additional effort is needed for this part of the discussion.

We thank the referee for his/her encouraging comment.

Finally, I ask the authors to more clearly mention in the paper the ongoing debate on the relative contribution of homogeneous and heterogeneous freezing, using also the reference mentioned in their answer (Barahona et al., 2017; Dietlicher et al., 2018).

We have added the following explanation:

"To the authors' knowledge, there is currently no observational constrain to the source of cloud ice in the mixed-phase regime. Namely, the frequency of ice clouds between 0°C and −42°C may be dominated by either convective ice detrainment or by in-situ freezing of cloud droplets. Overall, the relative contribution of heterogeneous and homogeneous freezing --- and the different INP types --- is still a matter of debate (Barahona et al., 2017; Dietlicher et al., 2018; Sullivan et al. 2017)."

**Main changes**

- Reorganization
  - The DARDAR and DARDAR-ALT product were removed from the main analysis. The comparison between GOCCP, DARDAR and ALT-DARDAR product was moved into the Appendix.
  - We moved the description of secondary products (2B-CLOUDSAT) to the appendix

- Deletions
  - Redundant figure descriptions similar to the figure captions were omitted (Fig. 3, 4, 6, 7, 8, 9 and 11)
  - Introduction was shortened.
  - Set. 2 — and particularly Sect. 2.4 — were shortened. Repetitions were removed, and the concept of volume gridbox is now better explained.
  - Sect.3 — particularly Sect. 3.3 — was simplified. The explanation about the filter for convective and precipitating clouds (remaining from the first versions) was omitted as the inclusion of these clouds do not change the results (due to their cloud cover; explained in text). Similarly, the weight on cloud volume fraction (old method) is also omitted, as it does not affect the results either. The weight with cloud volume fraction is only relevant as a weight for the meteorological parameters.
  - Omitted unused formulas
  - Omitted some repetitive text, specially about the percentiles
  - Acronyms: omitted $\beta$, $ATB_\perp$, and WBF
  - References to later parts are now omitted.

- Improvements
  - The figure about the day-to-day concept and the flowchart were improved and better explained in the text.
  - The day-to-day concept is now better explained, emphasizing that it is NOT the difference between neighbouring days.

[revised manuscript text omitted]

with

$$cvf_i \qquad\qquad\qquad\qquad\qquad\qquad\qquad\qquad = cvf_{i,altostratus} + cvf_{i,cirrus} + cvf_{i,altocumulus} + cvf_{i,stratocumulus} \qquad\qquad (3.3)$$

**Meteorological regimes**

Dust aerosol can produce or be accompanied by changes in atmospheric stability and  humidity. To disentangle such effects, we constrain the cloud environment  using the air relative humidity with respect to liquid and the tropospheric static stability. Depending on the isotherm to be studied, we use the lower troposphere static stability (LTSS) or the upper troposphere static stability (UTSS). These parameters are defined as:

$$\text{LTSS} = T_{700} \cdot [\frac{1000}{700}]^{R/C_p} - T_{sfc} \cdot [\frac{1000}{p_{sfc}}]^{R/C_p}$$

$$\text{UTSS} = T_{350} \cdot [\frac{1000}{350}]^{R/C_p} - T_{500} \cdot [\frac{1000}{p_{500}}]^{R/C_p}$$

[revised manuscript text omitted]